# BARYBIND: BINDING ALL MODALITIES VIA MULTIMODAL WASSERSTEIN BARYCENTER SPACE

## ABSTRACT

Multimodal joint representation, which aligns multiple modalities in a shared latent space, has emerged as the foundation of recent multimodal understanding models. To scale beyond two modalities, existing models typically treat a specific modality (e.g., text) as the anchor to bind other modalities via pairwise contrastive losses. However, the learned joint representation space tends to be suboptimal and imbalanced, as the modality-specific anchor may inherit the modality bias and insufficiently capture the modality-agnostic semantics and holistic geometric structures within multimodal data. In this work, we are motivated by the intuition that multimodal representations arise from different shifts from an underlying modality-agnostic representation space. Based on this, we present **BaryBind**, a multimodal framework that aligns modalities in the multimodal Wasserstein barycenter (WB) space, which inherently models a modality-agnostic distribution by minimizing the average of Wasserstein distances to all modalities. We further construct a barycenter polytope, whose volume serves as a geometric metric for quantifying $n$-modality alignment. This metric is integrated as a barycenter-anchored volumetric contrastive loss that contrasts the volumes of the $n$-dimensional polytopes, encouraging global alignment of non-anchor modalities to the barycenter while reducing inter-modality gaps. Extensive experiments show that BaryBind delivers more balanced zero-shot generalization performance in downstream tasks, e.g., cross-modal text/video retrieval and classification.

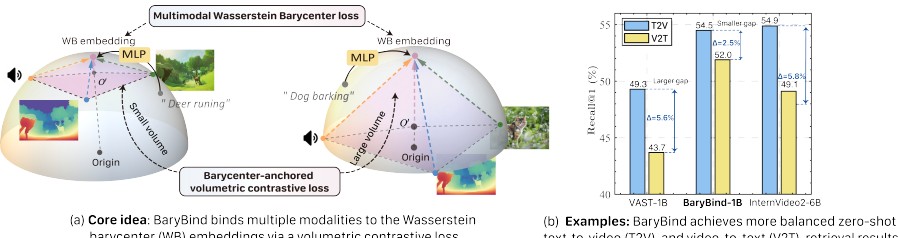

(a) **Core idea**: BaryBind binds multiple modalities to the Wasserstein barycenter (WB) embeddings via a volumetric contrastive loss

(b) **Examples:** BaryBind achieves more balanced zero-shot text-to-video (T2V) and video-to-text (V2T) retrieval results

Figure 1: **BaryBind binds multiple modalities to the Wasserstein barycenter (WB)**, which encodes modality-agnostic semantics by minimizing the average of Wasserstein distances to all modalities. By contrasting the barycenter polytope's volume, BaryBind achieves $n$-modality alignment to the WB space while preserving inter-modal interactions. Notably, it delivers more balanced retrieval results, reducing the T2V/V2T gap to 2.5% (vs. 5.6% for our baseline VAST Chen et al. (2023)).

## 1 INTRODUCTION

Multimodal learning (Baltrušaitis et al., 2018) seeks to integrate and process heterogeneous signals from multiple modalities (e.g., vision, language, audio, depth, *etc.*) to build a coherent perception of the surrounding world. Since multimodal data arise from heterogeneous observations of a shared underlying reality, recent multimodal learning methods (Radford et al., 2021; Jia et al., 2021) learn a shared representation space for representations from different modalities. In this field, the success of CLIP (Radford et al., 2021) in aligning unified vision–language representations via contrastive learning has sparked the adoption of contrastive losses as an appealing solution for multimodal representation learning. However, traditional contrastive losses, such as InfoNCE (Oord et al., 2018) and BYOL (Grill et al., 2020), are formulated in a pairwise fashion for $n = 2$ representation spaces typically arising from two modalities such as image–text (Radford et al., 2021; Jia et al., 2021) or audio–text (Guzhov et al., 2022; Elizalde et al., 2023) scenarios.

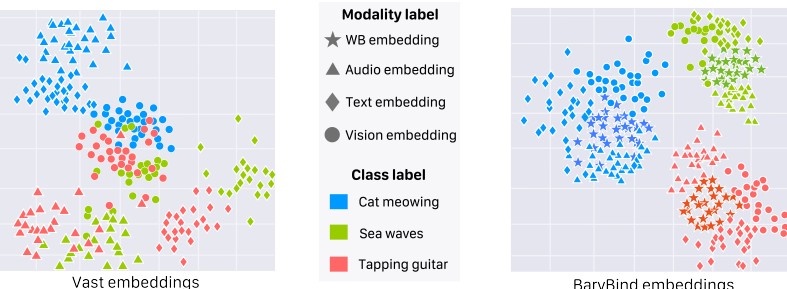

Figure 2: **The t-SNE comparison of embeddings on the zero-shot VGGSound dataset** (Chen et al., 2020). Compared to VAST (Chen et al., 2023), BaryBind induces a latent space where class clusters are clearly separated and multimodal embeddings are grouped around the WB embeddings.

While scaling to $n$ modalities ($n \geq 3$) poses unique challenges, a series of recent works (Chen et al., 2023; Zhu et al., 2024; Wang et al., 2025) originating from ImageBind (Girdhar et al., 2023) leverage the binding property of an modality-specific anchor (e.g., image or language) to align other modalities to the chosen anchor via pairwise losses. However, treating a specific modality as the alignment center can lead to sub-optimal shared space, as it inherits modality-specific biases while overlooking modality-agnostic semantics shared across all modalities. Consequently, the learned representations tend to be imbalanced, with a certain modality dominating the joint representation space (see the video/text retrieval results in Fig. 1). Moreover, although the pairwise losses align each modality to the anchor, they omit the correlations/interactions among non-anchor modalities, which may undermine the $n$-modality global alignment consistency.

To address these challenges, this work is motivated by the notion that *multimodal data are collected from heterogeneous sensors observing a shared underlying reality, thus the multimodal representations arise from different shifts from an underlying modality-agnostic representation space*. With this insight, we propose **BaryBind** that aligns multimodal representations in the multimodal Wassersiten Barycenter (WB) space, which inherently models a modality-agnostic distribution that minimizes the average of optimal transport (OT) distances to all modalities while capturing the OT-grounded geometry. This property inherently filters out modality-specific redundancy and results in a modality-agnostic barycenter space that reduces the divergence caused by multimodal domain shifts. Furthermore, we suggest a barycenter-anchored volumetric contrastive loss defined on the volume of a barycenter polytope, which quantifies the global alignment of $n$-dimensional multimodal data. The barycenter polytopes are spanned by the barycenter and non-anchor modality-to-barycenter gap vectors (see Fig. 1 (a)). By contrasting the polytope's volume, BaryBind binds the modalities to the modality-agnostic barycenter and reduces inter-modality gaps across modalities.

Specifically, we first propose a multimodal Wasserstein barycenter loss, which is optimized to seek the WB space for multimodal joint representation. This is achieved by training a lightweight map to filter out modality-specific biases, transforming the original modality-specific anchor to the WB that better approximates a modality-agnostic alignment anchor. (Fig. 1(a)) The barycenter is then leveraged to construct the $n$-dimensional barycenter polytope, whose volume serves as a measure of omni-modality alignment. Accordingly, we design a volumetric contrastive loss that encourages a smaller barycenter polytope volume for matched samples and a larger one for unmatched samples, thereby shaping a joint representation space that maintains the holistic geometry and inter-modal interactions within the multimodal data (Fig. 1 (a)). Our experiments show that BaryBind delivers competitive generalization performance in downstream tasks such as text/video retrieval and audio/video classification. Notably, BaryBind substantially alleviates the imbalance across modalities, reducing the T2V/V2T retrieval gap by 3.1% compared to VAST (Chen et al., 2023) (Fig. 1 (b)), and induces an embedding space where modalities of each class are consistently clustered around the WB embeddings, as presented in Fig. 2.

The main contributions of this paper are highlighted as follows:

• We propose BaryBind, a novel framework that aligns multimodal representations to the WB space. The WB space models a modality-agnostic distribution that captures the OT-grounded holistic geometry of multimodal data and inherently filters out modality-specific bias across modalities.

• We build a barycenter polytope with the barycenter and modality gap vectors. Its volume, serving as a global alignment metric, extends the insights of measuring $n$-modality alignment.

- Based on the polytope volume, we introduce a barycenter-anchored volumetric contrastive loss, which encourages global alignment to the barycenter while retaining inter-modal interactions.
- Extensive experiments show that BaryBind achieves more balanced representations and improves zero-shot generalization in downstream tasks such as text/video retrieval and audio/video classification. Particularly, BaryBind delivers decent scalability with increasing modality number.

## 2 RELATED WORKS

**Multimodal representation learning.** Multimodal representation learning seeks to align heterogeneous modalities into a shared semantic space. CLIP (Radford et al., 2021) initiates this paradigm with image–text contrastive learning, followed by audio extensions such as AudioCLIP (Guzhov et al., 2022), CLAP (Elizalde et al., 2023), and LAION-CLAP (Wu et al., 2023). WavCaps (Mei et al., 2024) constructs a large-scale audio captioning dataset to support audio–language retrieval. To scale beyond two modalities, ImageBind (Girdhar et al., 2023) introduces a pivot-based strategy, aligning each modality to the vision anchor. LanguageBind (Zhu et al., 2024), UniBind (Lyu et al., 2024), GRAM (Cicchetti et al., 2025b), and Triangle (Cicchetti et al., 2025a) extend alignment to language anchors, adopting unique volume-based contrastive losses. MiCo (Zhang et al., 2024) effectively expands modalities, data, and model size to learn unified representations. ViT-Lens (Lei et al., 2024) innovatively adapts a pretrained ViT via modality-specific "lenses" to establish a shared space. OmniBind (Wang et al., 2025) aligns pre-trained unimodal experts via pairwise losses. VAST (Chen et al., 2023) fuses modalities into a shared space but still relies on text-centered supervision. Despite their scalability, these approaches bind each modality to a specific modality (e.g., text) rather than modeling a modality-agnostic joint space. Consequently, they may suffer from modality-specific biases, which prevent them from learning a balanced joint space that captures the shared modality-agnostic semantics within multimodal data. Differently, BaryBind binds the modalities to the barycenter that encodes modality-agnostic semantics and adopts a barycenter-anchored volumetric contrastive loss to encourage the global alignment across $n$ modalities.

**Wasserstein barycenter.** The Wasserstein barycenter (Agueh & Carlier, 2011) defines an averaging distribution that minimizes the weighted sum of Wasserstein distances to input measures, preserving mass structure and OT-grounded geometry. This formulation has shown effectiveness on heterogeneous supports and has been applied in generative modeling (Cuturi & Doucet, 2014) and domain adaptation (Bonneel et al., 2015). Recent works estimate high-dimensional barycenters via deep dual formulations, including ICNN-based cycle-consistent models (Korotin et al., 2021), neural OT maps (Kolesov et al., 2024a; Tang et al., 2025), and energy-guided potentials (Kolesov et al., 2024b). We extend this perspective to construct a modality-agnostic joint space for multimodal alignment, bridging optimal transport theory with multimodal representation learning.

## 3 BARYBIND: BINDING VIA THE WASSERSTEIN BARYCENTER SPACE

In this section, we present BaryBind, a multimodal learning framework that aligns different modalities in the Wasserstein barycenter (WB) space. Our key insight is leveraging the inherent modality-agnostic nature of WB to build a joint space that mitigates divergence caused by modality-specific shifts. By aligning modalities to this space that encodes intrinsic modality-agnostic invariance, BaryBind potentially captures more balanced multimodal representations for downstream tasks.

Specifically, as shown in Fig. 3, BaryBind first constructs the multimodal WB space that aligns features from multimodal latent space via the multimodal WB loss (§3.2), in which an MLP is learned to transport the anchor features to the WB space as WB embeddings. We then construct a barycenter polytope defined by the WB embeddings and modality-to-barycenter gap vectors, whose volume quantifies the degree of $n$-modality alignment (§3.3). Based on the polytope, we introduce a barycenter-anchored volumetric contrastive loss, which encourages high-order multimodal alignment to the WB space while reducing inter-modality gaps (§3.4).

### 3.1 PRELIMINARIES

**Notation.** Let $\bar{K} = \{0, 1, \ldots, K\}$ for some $K \in \mathbb{N}$. For a sequence $e_0, \ldots, e_K$, we denote by $e_{0:K}$ the tuple $(e_0, \ldots, e_K)$. Let $\mathcal{X} \subset \mathbb{R}^d$, $\mathcal{Y} \subset \mathbb{R}^{d'}$, and $\mathcal{X}_k \subset \mathbb{R}^{d_k}$ be compact subsets of Euclidean space. Denote by $\mathcal{C}(\mathcal{X})$ the space of continuous real-valued functions on $\mathcal{X}$, and by $\mathcal{P}(\mathcal{X})$ the set of

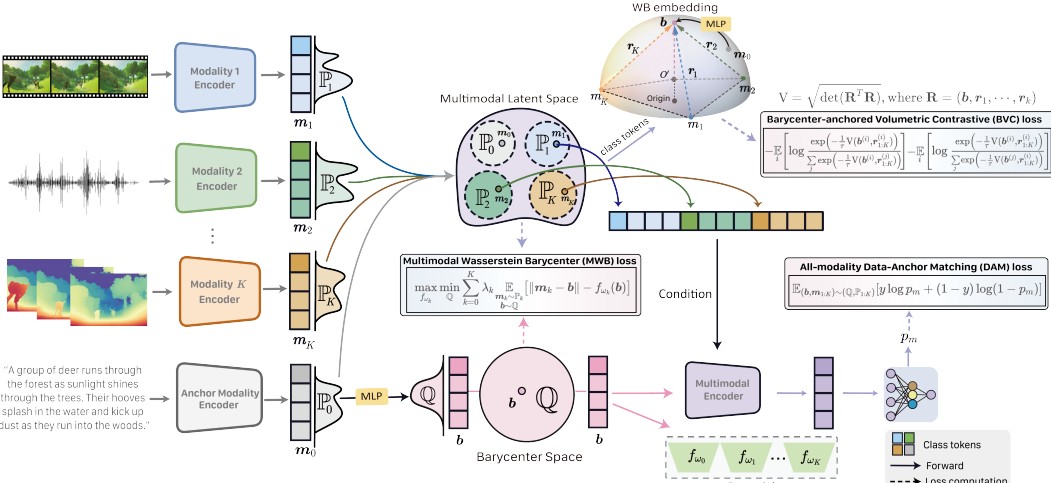

Figure 3: **The pipeline of BaryBind**. Given the multimodal embeddings, BaryBind constructs the WB space via the MWB loss, in which an MLP is learned to transform the anchor $\boldsymbol{m}_0$ into the WB embedding $\boldsymbol{b}$. The WB embedding acts as the new anchor and induces a barycenter polytope, whose volume is contrasted in the BVC loss to align other modalities. WB and non-anchor embeddings are then integrated in the multimodal encoder to predict whether the anchor and data are matched.

probability measures supported on $\mathcal{X}$. Given $\mathbb{P} \in \mathcal{P}(\mathcal{X})$ and $\mathbb{Q} \in \mathcal{P}(\mathcal{Y})$, we write $\Pi(\mathbb{P}, \mathbb{Q})$ for the set of transport plans between them, i.e., all joint distributions on $\mathcal{X} \times \mathcal{Y}$ whose marginals are $\mathbb{P}$ and $\mathbb{Q}$. The notation $\langle \cdot, \cdot \rangle$ denotes the cosine similarity that involves the normalization over features.

**Optimal transport.** Given two distributions $\mathbb{P} \in \mathcal{P}(\mathcal{X})$ and $\mathbb{Q} \in \mathcal{P}(\mathcal{Y})$, along with a cost function $c : \mathcal{X} \times \mathcal{Y} \to \mathbb{R}_+$, the classic optimal transport (OT) problem (Kantorovich, 1942) aims to find a joint distribution $\pi \in \Pi(\mathbb{P}, \mathbb{Q})$ that minimizes the expected transport cost:

$$\mathrm{OT}_c(\mathbb{P}, \mathbb{Q}) \triangleq \inf_{\pi \in \Pi(\mathbb{P}, \mathbb{Q})} \mathbb{E}_{(x,y) \sim \pi} \left[ c(x, y) \right]. \tag{1}$$

The specific choice of $c(x, y) = \|x - y\|$ yields $W(\mathbb{P}, \mathbb{Q}) = \inf_{\pi \in \Pi(\mathbb{P}, \mathbb{Q})} \mathbb{E}_{(x,y) \sim \pi} \|x - y\|$, known as the Earth-Mover or Wasserstein-1 distance. In this paper, we refer to it as the Wasserstein distance.

**Wasserstein barycenter (WB).** Given distributions $\mathbb{P}_k \in \mathcal{P}(\mathcal{X}_k)$ for $k \in \bar{K}$ and a vector $\lambda \in \mathbb{R}^{K+1}$ of non-negative weights $\lambda_k$ summing to 1, the WB problem seeks the distribution $\mathbb{Q}$ that minimizes the weighted sum of Wasserstein distances to the fixed marginals $\mathbb{P}_{0:K}$:

$$\inf_{\mathbb{Q} \in \mathcal{P}(\mathcal{Y})} \sum_{k=0}^{K} \lambda_k W(\mathbb{P}_k, \mathbb{Q}). \tag{2}$$

We apply the WB formulation to the multimodal latent space $\mathcal{M}$, where the encoded features of the $k$-th modality lie in a subspace $\mathcal{M}_k$ and follow a distribution $\mathbb{P}_k$. The barycenter inherently encodes modality-agnostic semantics as it models the "closest" distribution to all multimodal distributions.

## 3.2 MULTIMODAL WASSERSTEIN BARYCENTER SPACE

Let $\mathbb{P}_k$ be the distribution of encoded features $\boldsymbol{m}_k \in \mathcal{M}_k \subset \mathcal{M}$ for modality $k \in \bar{K}$, defined in the multimodal latent space $\mathcal{M} \subset \mathbb{R}^D$. In particular, $\boldsymbol{m}_0$ denote the anchor modality features. The WB space is defined as $\mathcal{M}_B := \mathrm{supp}(\mathbb{Q})$ where $\mathbb{Q}$ denotes the WB distribution and $\mathcal{M}_B$ contains the barycenter features $\boldsymbol{b}$. Given the distributions $\mathbb{P}_{0:K}$, our goal is to establish the barycenter space $(\mathcal{M}_B, \mathbb{Q})$ and use it as the joint representation space for multimodal alignment. Based on the WB formulation (2) over multimodal latent space, the multimodal WB problem can be written as

$$\mathcal{L}_{\mathrm{MWB}}^* = \inf_{\mathbb{Q} \in \mathcal{P}(\mathcal{M})} \sum_{k=0}^{K} \lambda_k W(\mathbb{P}_k, \mathbb{Q}). \tag{3}$$

However, directly optimizing (3) is highly intractable. To overcome this challenge, we deduce a new dual reformulation (see Appendix A.1), which leads to the following sup-inf objective:

**Proposition 1** (**Dual form of the multimodal WB problem**). *The infimum value $\mathcal{L}^*_{\mathrm{MWB}}$ of the multimodal WB problem (3) can be expressed as*

$$\mathcal{L}^*_{\mathrm{MWB}} = \sup_{\sum_k \lambda_k f_k = 0} \inf_{\mathbb{Q} \in \mathcal{P}(\mathcal{M}_B)} \sum_{k=0}^{K} \lambda_k \mathop{\mathbb{E}}_{\substack{\boldsymbol{m}_k \sim \mathbb{P}_k \\ \boldsymbol{b} \sim \mathbb{Q}}} \big[ \|\boldsymbol{m}_k - \boldsymbol{b}\| - f_k(\boldsymbol{b}) \big], \tag{4}$$

where the supremum is taken over all the dual potentials $f_k : \mathcal{M}_B \to \mathbb{R}$. We aim to learn the distribution $\mathbb{Q}$ by sampling the WB embeddings $\boldsymbol{b} = T(\boldsymbol{m}_0)$ via a trainable projection map $T_\theta(\cdot) : \mathcal{M}_0 \to \mathcal{M}_B$. Here, $\boldsymbol{m}_0$ represents the feature of anchor modality, typically selected as text, as it retains the broad semantics of multimodal data. We also perform an ablation study on the anchor selection, which is included in Appendix B. In practice, we simply parameterize $T_\theta$ using an MLP. To ensure the congruent constraint $\sum_k \lambda_k f_k = 0$ (Li et al., 2020), we parameterize the family of potentials $f_{\omega_k}$ as $g_{\omega_k} - \sum_{i=0}^{K} \lambda_i g_{\omega_i}$ with MLPs $g_{\omega_k} : \mathbb{R}^D \to \mathbb{R}$, which is a common trick used in (Li et al., 2020; Kolesov et al., 2024a;b).

**Multimodal Wasserstein Barycenter (MWB) loss**. With this parameterization, we rewrite (4) as a max-min objective of the MWB loss $\mathcal{L}_{\mathrm{MWB}}$, which can be optimized to compute the map $T_\theta$

$$\mathcal{L}^*_{\mathrm{MWB}}(\omega_{0:K}, \theta) \triangleq \max_{\omega_{0:K}} \min_{\theta} \sum_{k=0}^{K} \lambda_k \mathop{\mathbb{E}}_{\boldsymbol{m}_k \sim \mathbb{P}_k} \big[ \|\boldsymbol{m}_k - T_\theta(\boldsymbol{m}_0)\| - f_{\omega_k}(T_\theta(\boldsymbol{m}_0)) \big]. \tag{5}$$

To solve the problem (5), we train the networks $T_\theta$ and $f_{\omega_{0:K}}$ by alternately maximizing over $\omega_{0:K}$ and minimizing over $\theta$ in the MWB loss, in which we estimate the expectation using mini-batch data at each training step. Given the anchor modality feature $\boldsymbol{m}_0$, the WB embedding is then computed as $\boldsymbol{b} = T_\theta(\boldsymbol{m}_0)$, which will serve as the new anchor and other modalities are then aligned to it.

We establish the error bounds for the map $T$ with the following simplified notations:

$$\mathcal{F}(f_{0:K}, T) := \mathcal{L}_{\mathrm{MWB}}(f_{1:K}, T), \quad \mathcal{L}(f_{0:K}) := \inf_{T : \mathcal{M} \to \mathcal{M}_B} \mathcal{F}(f_{0:K}, T) \text{ and } \mathcal{L}^* := \mathcal{L}^*_{\mathrm{MWB}}. \tag{6}$$

**Theorem 3.1** (**Error analysis via duality gaps for the barycenter distribution**). *Let $C_k$ be any transport costs. Assume that the maps $\boldsymbol{b} \mapsto C_k(\boldsymbol{m}_k, \boldsymbol{b}) - \widehat{f}_k(\boldsymbol{b})$ are $\beta$-strongly convex for $\boldsymbol{m}_k \in \mathcal{M}_k$, $k \in \{0, \ldots, K\}$. Consider the duality gaps for an approximate solution $(\widehat{f}_{0:K}, \widehat{T})$:*

$$\mathcal{E}_1(\widehat{f}_{0:K}, \widehat{T}) \triangleq \mathcal{F}(\widehat{f}_{0:K}, \widehat{T}) - \mathcal{L}(\widehat{f}_{0:K}), \quad \mathcal{E}_2(\widehat{f}_{0:K}) \triangleq \mathcal{L}^* - \mathcal{L}(\widehat{f}_{0:K}),$$

*Then the following inequality holds:*

$$W^2\big(\widehat{T}_\# \mathbb{P}_0, \mathbb{Q}^*\big) \leq \frac{4}{\beta}\big(\mathcal{E}_1 + \mathcal{E}_2\big). \tag{7}$$

The proof is provided in the *Appendix* A.2. This theorem ensures that the Wasserstein distance between the estimated distribution $\widehat{T}_\# \mathbb{P}_0$ and the true barycenter $\mathbb{Q}^*$ is upper-bounded by the sum of these two errors. This establishes that, as both approximation and estimation errors decrease during training, the learned distribution converges toward the true WB in a distributional sense.

### 3.3 MEASURING $n$-MODALITY ALIGNMENT WITH BARYCENTER POLYTOPE VOLUME

To bind modalities to the WB space while preserving the global structures of $n$-dimensional (where $n = K + 1$ is the number of modalities) multimodal data, we introduce a geometric structure called the *barycenter polytope*. As illustrated in Fig. 3, the polytope takes the WB embedding $\boldsymbol{b}$ as the apex and is spanned by the vectors from the origin to $\boldsymbol{b}$ and the modality-to-barycenter gap vectors $\boldsymbol{r}_k = \boldsymbol{b} - \boldsymbol{m}_k$ for all non-anchor modalities $\boldsymbol{m}_k$ ($k \geq 1$). Owing to its unique composition, the polytope volume quantifies two aspects of multimodal features: (1) how closely non-anchor modalities align with the barycenter, and (2) the inter-modality discrepancy across modalities.

We use the volume of this barycenter polytope to measure $n$-modality alignment. Intuitively, a smaller volume indicates that the modalities are more tightly clustered around the WB embedding and more consistent with one another, suggesting stronger alignment. Conversely, a larger volume reflects greater distances from the barycenter and increased inter-modality discrepancy.

Given the WB embedding vector $\boldsymbol{b}$ (i.e., the vector from the origin to the barycenter) and gap vectors $\{\boldsymbol{r}_k\}_{k=1}^{K}$ (i.e., the vectors from each non-anchor modality to the barycenter), we define the matrix $\mathbf{R} = (\boldsymbol{b}, \boldsymbol{r}_1, \cdots, \boldsymbol{r}_K)$, representing the set of vectors spanning the barycenter polytope. Then the square of the polytope's volume can be computed according to Gantmacher (1959) as:

$$
\mathrm{V}^2(\boldsymbol{b}, \boldsymbol{r}_{0:K}) := \mathrm{V}^2(\boldsymbol{b}, \boldsymbol{r}_1, \cdots, \boldsymbol{r}_K) = \det(\mathbf{R}^T\mathbf{R}) = \begin{vmatrix} \langle \boldsymbol{b}, \boldsymbol{b} \rangle & \langle \boldsymbol{b}, \boldsymbol{r}_1 \rangle & \cdots & \langle \boldsymbol{b}, \boldsymbol{r}_K \rangle \\ \langle \boldsymbol{r}_1, \boldsymbol{b} \rangle & \langle \boldsymbol{r}_1, \boldsymbol{r}_1 \rangle & \cdots & \langle \boldsymbol{r}_1, \boldsymbol{r}_K \rangle \\ \vdots & \vdots & \ddots & \vdots \\ \langle \boldsymbol{r}_K, \boldsymbol{b} \rangle & \langle \boldsymbol{r}_K, \boldsymbol{r}_1 \rangle & \cdots & \langle \boldsymbol{r}_K, \boldsymbol{r}_K \rangle \end{vmatrix}. \quad (8)
$$

**Remark.** The volume offers a metric for measuring global alignment to the barycenter while retaining inter-modal interactions among non-anchor modalities, going beyond simple pairwise comparisons. The volume is computed as the determinant of a $n \times n$ matrix, where the number of modalities $n$ is typically small (e.g., 3 or 4) and much less than the embedding dimension $D$, introducing negligible computational overhead, which is further validated in our experiments.

### 3.4 BARYCENTER-BASED LOSS FUNCTIONS

To enforce tight and balanced multimodal binding in the Wasserstein barycenter space, we propose a barycenter-anchored volumetric contrastive (BVC) loss that leverages the volume of the barycenter polytope defined by the barycenter and modality-to-barycenter gap vectors.

Given multimodal inputs $\{\boldsymbol{m}_0^{(i)}, \boldsymbol{m}_1^{(i)}, \ldots, \boldsymbol{m}_K^{(i)}\}_{i=1}^{B}$, where $B$ is the batch size and $\boldsymbol{m}_0^{(i)}$ is the anchor modality. The barycenter is computed by $\boldsymbol{b}^{(i)} = T_\theta(\boldsymbol{m}_0^{(i)})$, and the modality-to-barycenter gap vectors are defined as $\boldsymbol{r}_k^{(i)} = \boldsymbol{b}^{(i)} - \boldsymbol{m}_k^{(i)}$ for $k = 1, \ldots, K$, aggregated as $\boldsymbol{r}_{0:K}^{(i)} = \{\boldsymbol{r}_1^{(i)}, \ldots, \boldsymbol{r}_K^{(i)}\}$. We treat each $(\boldsymbol{b}^{(i)}, \boldsymbol{r}_{0:K}^{(i)})$ as a *positive pair*, and construct two sets of *negative pairs*: (i) by fixing $\boldsymbol{b}^{(i)}$ and pairing it with $\boldsymbol{r}_{0:K}^{(j)}$ for $j \neq i$, and (ii) by fixing $\boldsymbol{r}_{0:K}^{(i)}$ and pairing it with $\boldsymbol{b}_j$ for $j \neq i$.

**Barycenter-anchored volumetric contrastive (BVC) loss**. The BVC loss contrasts small volumes for positive pairs against large volumes for negative ones, and is formulated as

$$
\mathcal{L}_{\mathrm{BVC}} = -\frac{1}{2}\, \mathbb{E}_i \left[ \log \frac{\exp\left(-\mathrm{V}(\boldsymbol{b}^{(i)}, \boldsymbol{r}_{0:K}^{(i)})/\tau\right)}{\sum_j \exp\left(-\mathrm{V}(\boldsymbol{b}^{(i)}, \boldsymbol{r}_{0:K}^{(j)})/\tau\right)} \right] - \frac{1}{2}\, \mathbb{E}_i \left[ \log \frac{\exp\left(-\mathrm{V}(\boldsymbol{b}^{(i)}, \boldsymbol{r}_{0:K}^{(i)})/\tau\right)}{\sum_j \exp\left(-\mathrm{V}(\boldsymbol{b}^{(j)}, \boldsymbol{r}_{0:K}^{(i)})/\tau\right)} \right], \quad (9)
$$

where $\mathrm{V}(\boldsymbol{b}, \boldsymbol{r}_{0:K})$ denotes the volume of the polytope formed by the barycenter and its modality gap vectors, whose square is computed as (8) and $\tau$ is the temperature parameter.

The BVC loss encourages non-anchor modalities to move closer to the common barycenter, ensuring tight binding around a shared modality-agnostic representation. Simultaneously, it provides a geometric guarantee that the inter-modality gap vectors converge, reducing discrepancies among modalities. To prevent trivial collapse of the polytope volume caused by shrinking vector magnitudes, we normalize each gap vector $\boldsymbol{r}_k$ to unit length. This normalization ensures that volume minimization effectively promotes alignment of vector directions rather than their norms, leading to a stable and balanced multimodal binding in the Wasserstein barycenter space.

**Data-anchor matching (DAM) loss.** We also introduce an auxiliary data-anchor matching (DAM) loss, which encourages the model to infer whether a pair of data and anchor is matched or not. This loss is commonly used in multimodal retrieval (Li et al., 2021b; Chen et al., 2023) to enforce fine-grained semantic alignment distinguishing matched and mismatched data pairs. To enable such matching, we can reuse the anchor-modality encoder as the multimodal encoder and integrate it with cross-attention layers, which take the WB embedding $\boldsymbol{b}$ as input and are conditioned on the unpooled concatenated embeddings $\boldsymbol{m}_{1:K}$ from non-anchor modalities. The output feature from the multimodal encoder is then passed through an two-layer MLP to produce binary predictions $p_m$. To construct informative in-batch negative pairs, we adopt a hard negative mining strategy following (Li et al., 2021a; Chen et al., 2023). The DAM loss is formulated as follows, where $y = 1$ if the barycenter anchor and non-anchor modalities are matched, and $y = 0$ otherwise:

$$
\mathcal{L}_{\mathrm{DAM}} = \mathbb{E}_{(\boldsymbol{b}, \boldsymbol{m}_{1:K}) \sim (\mathbb{Q}, \mathbb{P}_{1:K})}[y \log p_m(\boldsymbol{b}, \boldsymbol{m}_{1:K}) + (1 - y) \log(1 - p_m(\boldsymbol{b}, \boldsymbol{m}_{1:K}))] \quad (10)
$$

**The overall pre-training loss.** The overall pre-training objective $\mathcal{L}$ combines the three proposed losses, $\mathcal{L}_{\mathrm{MWB}}$, $\mathcal{L}_{\mathrm{BVC}}$, and $\mathcal{L}_{\mathrm{DAM}}$, as follows:

$$\mathcal{L} := \mathcal{L}_{\mathrm{MWB}} + \mathcal{L}_{\mathrm{BVC}} + \alpha \mathcal{L}_{\mathrm{DAM}}, \tag{11}$$

Particularly, the potentials $f_{\omega_{0:K}}$ in $\mathcal{L}_{\mathrm{MWB}}$ are optimized via maximization, and are trained in an alternating manner against the minimization of the remaining network parameters.

## 4 EXPERIMENTS

### 4.1 SETUP

We adopt VAST framework (Chen et al., 2023) as backbone, with BERT-B for text, BEATs for audio, and EVA-CLIP-ViT-G (Sun et al., 2023) for visual encoding. In total, the model comprises approximately 1B parameters. Unlike the VAST framework, we discard its modality fusion layers and introduce lightweight MLPs for barycenter optimization. We continue pretraining for one epoch based on VAST using our proposed loss functions on the VAST150k dataset (Chen et al., 2023), a subset of the VAST27M comprising 150k samples. The temperature factor $\tau$ is 0.07 and the trade-off parameter in (11) is set as $\alpha = 0.1$, following (Chen et al., 2023). The barycenter weights $\lambda_k$ ($k \in \bar{K}$) are uniformly set as $1/n$, where $n$ is the number of modalities.

We use benchmarks spanning diverse modalities to evaluate the BaryBind's capability in multimodal understanding across retrieval and classification tasks. These benchmarks include: (i) three-modality datasets such as DiDeMo (Anne Hendricks et al., 2017) and ActivityNet (Caba Heilbron et al., 2015), where video serves as the primary modality while audio and text provide auxiliary cues; (ii) four-modality datasets like MSR-VTT (Xu et al., 2016) and VATEX (Wang et al., 2019), covering video (V), audio (A), text (T), and subtitles (S); and (iii) audio-centered dataset VGGSound (Chen et al., 2020), where audio plays the dominant role and complementary information is also available in visual and textual forms. T-VAS denotes that the WB embedding is derived from the text modality, while the concatenated VAS embeddings serve as the condition for the multimodal encoder.

### 4.2 COMPARISON WITH STATE-OF-THE-ART METHODS

**Zero-shot video/audio classification.** We first evaluate BaryBind on VGGSound5K to assess its multimodal understanding ability in the zero-shot setting, particularly when jointly modeling video and audio signals. As shown in Tab. 1, BaryBind achieves the best top-1 and top-5 accuracy across all modality configurations, reaching 55.6% Acc@1 and 83.4 % Acc@5 under the A+V setting. It significantly outperforms the VAST baseline (48.1% / 79.6%) and other strong competitors such as OmniBind (45.4% / 73.2%). This demonstrates its strong generalization ability across modalities. The substantial gains with A+V highlight the enhanced performance of cross-modal understanding, enabled by the BVC loss design which inherently reduces inter-modality gaps and preserves the global geometry of multi-

Table 1: **Zero-shot video/audio classification** results on VGGSound5K. Results from our baseline VAST and BaryBind are highlighted accordingly.

| Method | Modality | Acc@1 | Acc@5 |
|---|---|---|---|
| ImageBind (Girdhar et al., 2023) | A | 31.6 | 58.7 |
| ImageBind (Girdhar et al., 2023) | V | 37.9 | 65.9 |
| LanguageBind (Zhu et al., 2024) | A | 34.1 | 62.8 |
| LanguageBind (Zhu et al., 2024) | V | 39.6 | 64.5 |
| GRAM (Cicchetti et al., 2025b) | V | 43.1 | 71.8 |
| GRAM (Cicchetti et al., 2025b) | A+V | 42.3 | 74.5 |
| Triangle (Cicchetti et al., 2025a) | A+V | 44.8 | 80.0 |
| OmniBind (Wang et al., 2025) | A | 41.7 | 70.8 |
| OmniBind (Wang et al., 2025) | V | 45.4 | 73.2 |
| OmniBind (Wang et al., 2025) | A+V | 46.2 | 76.2 |
| VAST (Chen et al., 2023) | A | 40.3 | 71.7 |
| VAST (Chen et al., 2023) | V | 46.3 | 72.7 |
| VAST (Chen et al., 2023) | A+V | 48.1 | 79.6 |
| BaryBind (Ours) | A | 45.7 | 75.2 |
| BaryBind (Ours) | V | 48.3 | 76.4 |
| BaryBind (Ours) | A+V | **55.6** | **83.4** |

modal data by aligning multimodal representation based on the barycenter polytope volume.

On the other hand, BaryBind improves the performance of the weaker audio-only modality classification (A) from 40.3% (VAST) to 45.7% for Acc@1, indicating its ability to alleviate under-optimization of weaker modalities. These results confirm that BaryBind learns more balanced multimodal representations by aligning modalities via our barycenter-based binding strategy.

**Cross-modal retrieval.** We evaluate BaryBind on zero-shot text-to-video (T2V) and video-to-text (V2T) retrieval across four benchmarks. As shown in Tab. 2, BaryBind consistently achieves state-of-the-art results under all modality configurations. For instance, under the T-VA setting, it achieves

Table 2: **Zero-shot** text-to-video (T2V) and video-to-text (V2T) retrieval results in terms of Recall at 1 score (R@1). Results from our baseline VAST and BaryBind are highlighted accordingly.

| Methods | Modality | MSR-VTT | | DiDeMo | | ActivityNet | | VATEX | |
|---|---|---|---|---|---|---|---|---|---|
| | | T2V | V2T | T2V | V2T | T2V | V2T | T2V | V2T |
| VideoCoCa (Yan et al., 2022) | T-V | 34.3 | 64.7 | - | - | 34.5 | 33.0 | 53.2 | 73.6 |
| X-CLIP (Ma et al., 2022) | T-V | 46.1 | 46.8 | 45.2 | 42.3 | 44.3 | 42.6 | - | - |
| ImageBind (Girdhar et al., 2023) | T-V | 36.8 | - | - | - | - | - | - | - |
| ViCLIP (Wang et al., 2024c) | T-V | 42.4 | 41.3 | 18.4 | 27.9 | 15.1 | 24.0 | - | - |
| VideoPrism-b (Zhao et al., 2024) | T-V | 51.4 | 50.2 | - | - | 49.6 | 47.9 | 62.5 | 77.1 |
| LanguageBind (Zhu et al., 2024) | T-V | 44.8 | 40.9 | 39.9 | 39.8 | 41.0 | 39.1 | - | - |
| InternVL (Chen et al., 2024) | T-V | 46.3 | 42.4 | 43.7 | 42.2 | 45.1 | 42.4 | 66.8 | 69.3 |
| OmniBind (Wang et al., 2025) | T-V | 47.4 | 45.2 | 43.5 | 42.6 | 44.3 | 40.8 | - | - |
| NarVid (Hur et al., 2025) | T-V | 51.8 | 50.3 | 52.4 | 50.5 | 51.8 | 46.6 | 73.8 | 76.3 |
| Video-ColBERT (Reddy et al., 2025) | T-V | 51.9 | 48.8 | 51.7 | 50.1 | 52.7 | 47.8 | 72.4 | 73.7 |
| GRAM (Cicchetti et al., 2025b) | T-VAS | 54.2 | 51.6 | - | - | - | - | 83.2 | 81.9 |
| VAST (Chen et al., 2023) | T-VA | 49.3 | 43.7 | 49.5 | 48.2 | 51.4 | 46.8 | 80.0 | 77.3 |
| VAST (Chen et al., 2023) | T-VAS | 50.9 | 47.9 | - | - | - | - | 82.1 | 78.7 |
| BaryBind (Ours) | T-V | 53.4 | 51.3 | 54.3 | 52.7 | 59.3 | 51.6 | 82.3 | 79.8 |
| BaryBind (Ours) | T-VA | 54.5 | 52.0 | **55.3** | **52.8** | **59.6** | **53.9** | 84.2 | 81.3 |
| BaryBind (Ours) | T-VAS | **56.3** | **53.6** | - | - | - | - | **84.6** | **83.5** |
| InternVideo2-6B (Wang et al., 2024d) | T-VA | 54.9 | 49.1 | 55.7 | 51.6 | 61.2 | 52.8 | 82.7 | 76.4 |

Table 3: **Finetuning** text-to-video (T2V) and video-to-text (V2T) retrieval results in terms of Recall at 1 score (R@1). Results from our baseline VAST and BaryBind are highlighted accordingly.

| Methods | Modality | MSR-VTT | | DiDeMo | | ActivityNet | | VATEX | |
|---|---|---|---|---|---|---|---|---|---|
| | | T2V | V2T | T2V | V2T | T2V | V2T | T2V | V2T |
| CLIP4Clip (Luo et al., 2021) | T-V | 45.6 | 45.9 | 43.0 | 43.6 | 40.3 | 41.6 | 63.0 | 78.3 |
| InternVideo-L (Wang et al., 2022) | T-V | 53.1 | 54.4 | 57.9 | 59.1 | 62.2 | 62.8 | 69.8 | 80.6 |
| HiTeA (Ye et al., 2022) | T-V | 46.8 | - | 56.5 | - | - | - | - | - |
| mPLUG-2 (Xu et al., 2023) | T-V | 53.1 | - | 56.4 | - | - | - | - | - |
| TEFAL (Ibrahimi et al., 2023) | T-VA | 52.0 | - | - | - | - | - | 61.0 | - |
| ViCLIP (Wang et al., 2024c) | T-V | 52.5 | 51.8 | 49.4 | 50.2 | 49.8 | 48.1 | - | - |
| T-MASS (Wang et al., 2024a) | T-VA | 52.7 | - | 53.3 | - | - | - | 65.6 | - |
| VALOR-L (Liu et al., 2024) | T-VAS | 54.4 | - | 57.6 | - | 63.4 | - | 76.9 | - |
| VideoCLIP-XL (Wang et al., 2024b) | T-V | 54.6 | 54.0 | 62.3 | 62.7 | 58.4 | 59.2 | - | - |
| TempMe (Shen et al., 2025) | T-V | 49.0 | 47.6 | 48.0 | 48.4 | 44.9 | 45.3 | 69.6 | 71.8 |
| VAST (Chen et al., 2023) | T-VA | 55.8 | 57.6 | 65.6 | 62.0 | 68.8 | 66.7 | 86.9 | 84.1 |
| VAST (Chen et al., 2023) | T-VAS | 56.6 | 57.6 | - | - | - | - | 87.5 | 84.0 |
| BaryBind (Ours) | T-V | 57.4 | 57.8 | 67.2 | 64.6 | 67.3 | 65.2 | 85.0 | 82.4 |
| BaryBind (Ours) | T-VA | 60.3 | 60.8 | **68.5** | **64.4** | **72.1** | **69.4** | 87.4 | **84.8** |
| BaryBind (Ours) | T-VAS | **64.6** | **65.2** | - | - | - | - | **88.4** | 84.6 |

55.3 and 52.8 R@1 on DiDeMo (T2V/V2T), outperforming all prior methods. With T-VAS, Bary-Bind further sets new records on MSR-VTT (56.3/53.6) and VATEX (84.6/83.5), demonstrating strong retrieval performance across both directions. In addition to these overall gains, BaryBind significantly reduces the T2V/V2T performance gap (e.g., 5.7 on ActivityNet vs. 9.6 for the VAST baseline), indicating improved bidirectional alignment. This suggests that BaryBind learns more generalizable and balanced cross-modal representations, which can be attributed to the WB binding mechanism that geometrically unifies diverse modalities into a modality-agnostic latent space.

We present the fine-tuning results in Tab. 3, which shows BaryBind maintains its advantage, out-performing prior models such as InternVideo, VideoCLIP-XL, and VAST. For instance, our T-VA setting achieves 60.3 R@1 on MSR-VTT and 72.1 on ActivityNet for T2V, surpassing VAST by +4.5 and +2.3 points, respectively. The performance further improves under T-VAS, where Bary-Bind reaches 64.6 R@1 on MSR-VTT, a +7.4 point gain over VAST, indicating that our binding strategy generalizes well with additional modalities and scales effectively with fine-tuning.

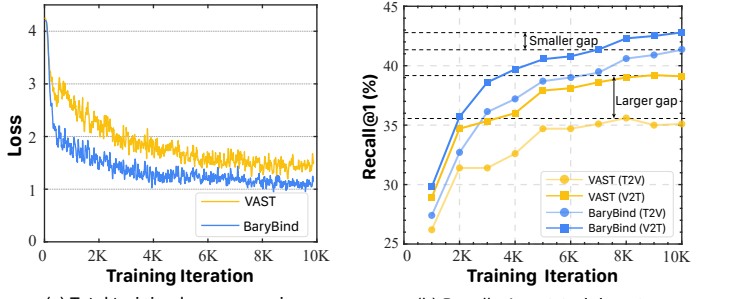

Figure 4: **Training dynamics comparison with cosine-based VAST:** (a) Total training loss showing faster convergence for BaryBind. (b) Recall@1 (R@1) evolution on MSR-VTT under T2V and V2T retrieval tasks. BaryBind achieves higher retrieval accuracy and better modal balance (smaller R@1 gap). (c) V2T/T2V retrieval gaps throughout training.

## 4.3 ABLATION STUDIES

**Training dynamics comparison.** We compare the training (from scratch) behaviors of BaryBind and the baseline VAST to assess the impact of our binding strategy and loss design. As shown in Fig. 4, BaryBind exhibits consistently superior training dynamics. In (a), the total loss drops faster and stabilizes earlier, suggesting that our MWB and BVC losses facilitate more efficient optimization. In (b), BaryBind consistently achieves higher R@1 accuracy for both T2V and V2T retrieval, while maintaining a smaller performance gap between the two directions, suggesting more symmetric multimodal representations. (c) demonstrates that BaryBind rapidly reduces the gap between V2T and T2V performance, converging toward a balanced retrieval behavior. This improvement stems from the joint effect of MWB and the BVC loss, which together promote consistent alignment of modality embeddings toward a unified barycenter. Such training behavior reflects the effectiveness of our barycentric modeling in encouraging modality-agnostic representation learning.

Table 4: **Ablation study on loss functions.** We report top-1 classification accuracy (Acc@1) on VGGSound5K and top-1 retrieval recall at 1 score (R@1) on MSR-VTT. Key improvements from MWB, BVC, and their combination are highlighted correspondingly.

| Loss function components | | | | Classification | | | Retrieval | |
|---|---|---|---|---|---|---|---|---|
| | | | | VGGSound | | | MSR-VTT | |
| TV+TA CL | MWB | BVC | DAM | A | V | V+A | T2V | V2T |
| ✓ | ✗ | ✗ | ✗ | 38.1 | 44.5 | 46.8 | 46.8 | 40.1 |
| ✓ | ✗ | ✗ | ✓ | 40.3 | 46.3 | 48.1 | 49.3 | 43.7 |
| ✓ | ✓ | ✗ | ✗ | 43.6 | 45.6 | 47.6 | 48.8 | 46.2 |
| ✓ | ✓ | ✗ | ✓ | 44.3 | 46.8 | 49.8 | 49.7 | 48.3 |
| ✗ | ✗ | ✓ | ✗ | 42.9 | 46.3 | 50.3 | 50.6 | 46.4 |
| ✗ | ✗ | ✓ | ✓ | 43.4 | 47.2 | 52.6 | 51.5 | 46.8 |
| ✗ | ✓ | ✓ | ✗ | 45.2 | 47.8 | 54.2 | 53.2 | 50.6 |
| ✗ | ✓ | ✓ | ✓ | **45.7** | **48.3** | **55.6** | **54.5** | **52.0** |

**Comparison of loss functions.** To evaluate the contribution of each loss component, we conduct an ablation study on VGGSound (Chen et al., 2020) for audio classification and MSR-VTT (Xu et al., 2016) for text–video retrieval under a tri-modal setting (text, video, audio). The baseline adopts pairwise cosine contrastive losses (TV+TA CL), while VAST (Chen et al., 2023) additionally includes the data-anchor matching (DAM) loss. As summarized in Tab. 4, introducing the multimodal Wasserstein barycenter (MWB) loss consistently improves performance across tasks—for example, boosting audio-only classification from 40.3% to 44.3% and V2T retrieval from 43.7% to 48.3%. MWB constructs a barycentric semantic anchor to filter out modality-specific biases from the original anchor (e.g., text), leading to a more balanced and modality-agnostic anchor. The barycenter volume contrastive (BVC) loss further enhances global geometric consistency by preserving the relative structures among modalities, enabling full-modality gains such as 55.6% Acc@1 in audio classification and 54.5/52.0 R@1 for T2V/V2T retrieval. Complementarily, the DAM loss leverages

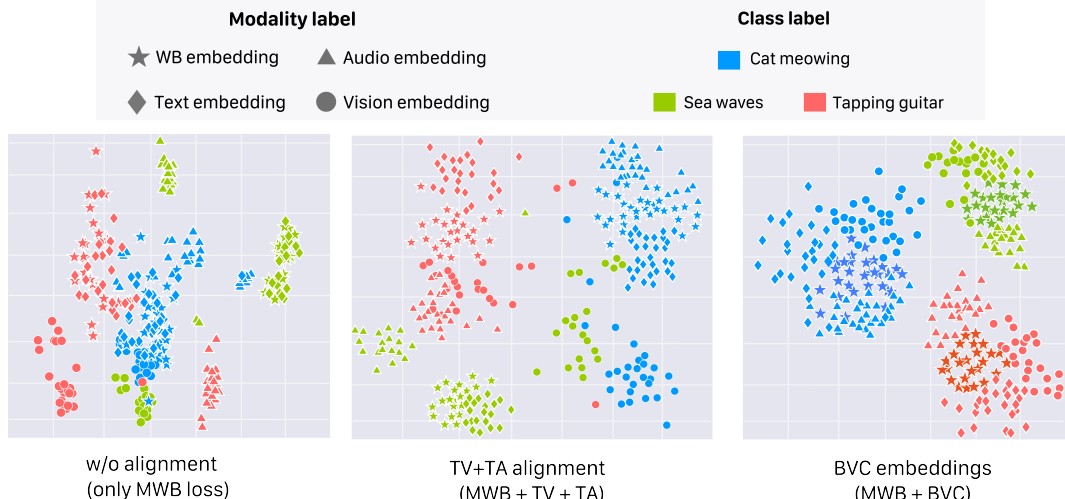

Figure 5: **Visualization of multimodal before and after alignment.** Embeddings before (left) and after (right) applying the proposed BVC loss. The BVC loss promotes convergence of modality-specific embeddings toward the Wasserstein barycenter, reducing inter-modality gaps and forming compact, modality-agnostic clusters, which highlights improved multimodal alignment.

instance-level supervision to distinguish matched and mismatched cross-modal pairs. Overall, the combination of MWB, BVC, and DAM facilitates a collaborative effect by aligning all modalities toward a shared semantic barycenter, yielding more balanced and generalizable multimodal representations.

**Visualization of multimodal embeddings before and after alignment**. To intuitively illustrate the effect of our barycenter-based alignment, we visualize the embeddings of three categories (cat meowing, sea waves, tapping guitar) across audio, text, and vision modalities from VGGSound (Chen et al., 2020), as shown in Figure 5. Without alignment (left), embeddings of different modalities are scattered, exhibiting large inter-modality gaps even within the same class. Incorporating pairwise alignment (middle) improves class clustering but still reveals modality-specific separations. In contrast, with our proposed BVC loss (right), embeddings of all modalities converge tightly around their class-wise Wasserstein barycenters, leading to compact intra-class structures and reduced inter-modality discrepancies. This clearly demonstrates that the volumetric constraint effectively promotes modality-agnostic alignment around a unified barycenter space.

## 5 CONCLUSION

We introduced BaryBind, a novel multimodal learning framework that aligns multiple modalities to a shared multimodal Wasserstein barycenter space. Unlike traditional anchor-based alignment strategies, BaryBind leverages the barycenter to model a modality-agnostic semantic distribution, providing a principled and geometry-aware alignment target. By constructing a barycenter polytope and leveraging its volume as a global alignment metric, BaryBind captures higher-order interactions among modalities and quantifies alignment quality beyond pairwise similarities. The proposed volumetric contrastive loss further encourages all modalities to converge toward the barycenter while preserving inter-modal structure. Extensive experiments on retrieval and classification tasks demonstrate that BaryBind learns more balanced and generalizable multimodal representations, outperforming state-of-the-art approaches across diverse benchmarks. We hope this work opens new directions for scalable multimodal learning and inspires the development of more interpretable and geometry-grounded multimodal methods.

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

## A  THEORETICAL RESULTS

### A.1  PROOF OF THEOREM 1

*Proof.* Beyond the WB problem, we deduce based on the duality of with general costs $C_k(x, y)$. In this sense, we write the dual reformulation of the multimodal OT barycenter problem (3):

$$\mathcal{L}^* = \inf_{\mathbb{Q} \in \mathcal{P}(\mathcal{M}_B)} \sup_{f_0, \dots, f_k \in \mathcal{C}(\mathcal{M}_B)} \underbrace{\underbrace{\sum_{k=1}^{K} \lambda_k \left\{ \int_{\mathcal{M}_k} f_k^{C_k}(\boldsymbol{m}_k) \mathrm{d}\mathbb{P}_k(\boldsymbol{m}_k) + \int_{\mathcal{M}_B} f_k(\boldsymbol{b}) \mathrm{d}\mathbb{Q}(\boldsymbol{b}) \right\}}_{\triangleq \widetilde{\mathcal{F}}(\mathbb{Q}, f_{0:K})}}_{\triangleq \mathcal{L}(f_{0:K})}. \quad (12)$$

where

$$f_k^{C_k}(\boldsymbol{m}_k) = \inf_{b \in \mathcal{M}_B} [C_k(\boldsymbol{m}_k, \boldsymbol{b}) - f_k(\boldsymbol{b})]. \quad (13)$$

We denote the expression under the $\inf$ and $\inf \sup$ in (12) as functionals $\mathcal{L} : \mathcal{C}(\mathcal{M}_B)^K$ and $\widetilde{\mathcal{F}} : \mathcal{P}(\mathcal{M}_B) \times \mathcal{C}(\mathcal{M}_B)^K$, respectively. For simplicity, we also introduce the following notation

$$\bar{f} \triangleq \sum_{k=1}^{K} \lambda_k f_k \quad \text{and} \quad M \triangleq \inf_{\boldsymbol{b} \in \mathcal{M}_B} \bar{f}(\boldsymbol{b}) = \inf_{\mathbb{Q} \in \mathcal{P}(\mathcal{M}_B)} \int_{\mathcal{M}_B} \bar{f}(\boldsymbol{b}) d\mathbb{Q}(\boldsymbol{b}), \quad (14)$$

where the equality follows from two fundamental observations: (a) $M \leq \int \bar{f}(\boldsymbol{b}) \, d\mathbb{Q}(\boldsymbol{b})$ for any $\mathbb{Q} \in \mathcal{P}(\mathcal{M}_B)$, and (b) $\bar{f}(\boldsymbol{b}) = \int \bar{f}(\boldsymbol{b}') \, d\delta_{\boldsymbol{b}}(\boldsymbol{b}')$ where $\delta_{\boldsymbol{b}}$ represents the Dirac mass at $\boldsymbol{b} \in \mathcal{M}_B$.

Firstly, due to the compactness of $\mathcal{M}_B$, the space $\mathcal{P}(\mathcal{M}_B)$ is compact with respect to the weak topology. For fixed potentials $f_{0:K} \in \mathcal{P}(\mathcal{M}_B)^K$ we have that $\widetilde{\mathcal{F}}(\cdot, f_{0:K})$ is linear, convex and continuous. Secondly, for a fixed $\mathbb{Q}$, the functional $\widetilde{\mathcal{F}}(\mathbb{Q}, \cdot)$ is a concave due to the concavity of $C$-transform. These properties enable the application of Sion's minimax theorem (Sion (1958), Theorem 3.4), which allows the interchange of the $\sup$ and $\inf$ in (12). Thus with (14) we obtain

$$\mathcal{L}^* = \sup_{f_0, \dots, f_K \in \mathcal{C}(\mathcal{M}_B)} \inf_{\mathbb{Q} \in \mathcal{P}(\mathcal{M}_B)} \sum_{k=1}^{K} \lambda_k \left\{ \int_{\mathcal{M}_k} f_k^{C_k}(\boldsymbol{m}_k) \mathrm{d}\mathbb{P}_k(\boldsymbol{m}_k) + \int_{\mathcal{M}_B} f_k(\boldsymbol{b}) \mathrm{d}\mathbb{Q}(\boldsymbol{b}) \right\}$$

$$= \sup_{f_0, \dots, f_K \in \mathcal{C}(\mathcal{M}_B)} \left\{ \sum_{k=1}^{K} \lambda_k \int_{\mathcal{M}_k} f_k^{C_k}(\boldsymbol{m}_k) \mathrm{d}\mathbb{P}_k(\boldsymbol{m}_k) + \inf_{\mathbb{Q} \in \mathcal{P}(\mathcal{M}_B)} \int_{\mathcal{M}_B} \bar{f}(\boldsymbol{b}) \mathrm{d}\mathbb{Q}(\boldsymbol{b}) \right\}$$

$$= \sup_{f_0, \dots, f_K \in \mathcal{C}(\mathcal{M}_B)} \underbrace{\left\{ \sum_{k=1}^{K} \lambda_k \int_{\mathcal{M}_k} f_k^{C_k}(\boldsymbol{m}_k) \mathrm{d}\mathbb{P}_k(\boldsymbol{m}_k) + \inf_{\boldsymbol{b} \in \mathcal{M}_B} \bar{f}(\boldsymbol{b}) \right\}}_{\triangleq \widetilde{\mathcal{L}}(f_{0:K})}. \quad (15)$$

Now we show that the $\sup$ in (15) can be restricted to potentials $\tilde{f}_{0:K}$ which satisfy the congruence condition $\sum_{k=1}^{K} \lambda_k \tilde{f}_k = 0$. It is enough to show that for every tuple $f_{0:K}$ there exists a congruent tuple $\tilde{f}_{0:K} \in \mathcal{C}(\mathcal{M}_B)^K$ such that $\widetilde{\mathcal{L}}(\tilde{f}_{0:K}) \geq \widetilde{\mathcal{L}}(f_{0:K})$.

For this, we consider the congruent potentials given any tuple $f_{0:K}$

$$(\tilde{f}_0, \cdots, \tilde{f}_K) = \left( f_0, \cdots, f_{K-1}, f_K - \frac{\bar{f}}{\lambda_K} \right). \quad (16)$$

Since $\tilde{M} \triangleq \inf_{\boldsymbol{b}\in\mathcal{M}_B} \sum_{k=1}^{K} \lambda_k \tilde{f}_k = 0$, we obtain

$$\widetilde{\mathcal{L}}(\tilde{f}_{0:K}) - \widetilde{\mathcal{L}}(f_{0:K}) = \lambda_K \int_{\mathcal{M}_k} \left( \tilde{f}_K^{C_K}(\boldsymbol{m}_K) - f_K^{C_K}(\boldsymbol{m}_K) \right) d\mathbb{P}_K(\boldsymbol{m}_K) - M$$

$$= \lambda_K \int_{\mathcal{M}_k} \left[ \left( f_K - \frac{\bar{f}}{\lambda_K} \right)^{C_K} (\boldsymbol{m}_K) - f_K^{C_K}(\boldsymbol{m}_K) \right] d\mathbb{P}_K(\boldsymbol{m}_K) - M$$

$$\geq \lambda_K \int_{\mathcal{M}_k} \left[ \left( f_K - \frac{M}{\lambda_K} \right)^{C_K} (\boldsymbol{m}_K) - f_K^{C_K}(\boldsymbol{m}_K) \right] d\mathbb{P}_K(\boldsymbol{m}_K) - M$$

$$= \lambda_K \int_{\mathcal{M}_k} \frac{M}{\lambda_K} d\mathbb{P}_K(\boldsymbol{m}_K) - M = 0, \tag{17}$$

where the first inequality arises from the monotonicity of the $C$-transform, along with the fact $\bar{f}_K = f_K - \frac{\bar{f}}{\lambda_K} \leq f_K - \frac{M}{\lambda_k}$. The last equality follows from the definition of $C$-transform.

Finally, since $\widetilde{\mathcal{L}}(f_{0:K}) = \mathcal{L}(f_{0:K})$ for congruent potentials $f_{0:K}$, with (13) we obtain

$$\mathcal{L}^* = \sup_{\sum_k \lambda_k f_k = 0} \mathcal{L}(f_{0:K}) = \sup_{\sum_k \lambda_k f_k = 0} \sum_{k=1}^{K} \lambda_k \int_{\mathcal{M}_k} f_k^{C_k}(\boldsymbol{m}_k) d\mathbb{P}_k(\boldsymbol{m}_k),$$

$$= \sup_{\sum_k \lambda_k f_k = 0} \sum_{k=1}^{K} \lambda_k \inf_{\boldsymbol{b}\in\mathcal{M}_B} \int_{\mathcal{M}_k} \left[ C_k(\boldsymbol{m}_k - \boldsymbol{b}) - f_k(\boldsymbol{b}) \right] d\mathbb{P}_k(\boldsymbol{m}_k). \tag{18}$$

In practice, we replace each integral with an empirical expectation over the distribution $\mathbb{P}_k$, and adopt $C_k(\boldsymbol{m}_k - \boldsymbol{b}) = \|\boldsymbol{m}_k - \boldsymbol{b}\|$, which corresponds to the Wasserstein distance between the modality feature and the barycenter. We exchange the summation and the infimum since $\boldsymbol{b}$ is shared across all terms and the objective is linear. We then sample the barycenter $\boldsymbol{b}$ from a distribution $\mathbb{Q} \in \mathcal{P}(\mathcal{M}_B)$, and express the integral as an expectation over both $\boldsymbol{m}_k \sim \mathbb{P}_k$ and $\boldsymbol{b} \sim \mathbb{Q}$:

$$\mathcal{L}^* = \sup_{\sum_k \lambda_k f_k = 0} \inf_{\mathbb{Q}\in\mathcal{P}(\mathcal{M}_B)} \sum_{k=0}^{K} \lambda_k \mathop{\mathbb{E}}_{\substack{\boldsymbol{m}_k\sim\mathbb{P}_k \\ \boldsymbol{b}\sim\mathbb{Q}}} \left[ \|\boldsymbol{m}_k - \boldsymbol{b}\| - f_k(\boldsymbol{b}) \right], \tag{19}$$

which completes the proof. $\qquad\square$

### A.2 PROOF OF THEOREM 3.1

*Proof.* Let $\mathbb{P}_{0:K}$ denote the joint distribution over the multimodal tuples $(\boldsymbol{m}_0, \ldots, \boldsymbol{m}_K) =: \boldsymbol{m}_{0:K}$, where $\boldsymbol{m}_k \in \mathcal{M}_k$. The barycenter map is a function $T : \mathcal{M}_0 \to \mathcal{M}_B$ that acts on the anchor modality. For a set of potential functions $f_{0:K}$ and a map $T$, we rewrite the total cost functional $\mathcal{F}$ and its corresponding minimal cost functional $\mathcal{L}$ as follows:

$$\mathcal{F}(f_{0:K}, T) \triangleq \mathbb{E}_{\boldsymbol{m}_{0:K}\sim\mathbb{P}_{0:K}} \left[ \sum_{k=0}^{K} \lambda_k \left( C_k(\boldsymbol{m}_k, T(\boldsymbol{m}_0)) - f_k(T(\boldsymbol{m}_0)) \right) \right], \tag{20}$$

$$\mathcal{L}(f_{0:K}) \triangleq \inf_{T:\mathcal{M}_0\to\mathcal{M}_B} \mathcal{F}(f_{0:K}, T). \tag{21}$$

The minimizer of functional $\mathcal{F}$ is denoted as:

$$T^f \in \mathop{\arg\inf}_{T:\mathcal{M}\to\mathcal{M}_B} \mathcal{F}(\widehat{f}_{0:K}, T). \tag{22}$$

Given $T^f : \mathcal{M}_0 \to \mathcal{M}_B$, the functional $\mathcal{L}$ in (21) can be written as:

$$\mathcal{L}(\widehat{f}_{0:K}) = \mathcal{F}(\widehat{f}_{0:k}, T^f). \tag{23}$$

We can observe that the first gap $\mathcal{E}_1$ is the difference between (20) and (23):

$$\mathcal{E}_1(\widehat{f}_{0:K}, \widehat{T}) = \mathcal{F}(\widehat{f}_{0:K}, \widehat{T}) - \mathcal{L}(\widehat{f}_{0:K}) = \mathcal{F}(\widehat{f}_{0:K}, \widehat{T}) - \mathcal{F}(\widehat{f}_{0:K}, T^f). \tag{24}$$

Before looking into the second gap $\mathcal{E}_2$, we recall the optimal value $\mathcal{L}^*$ of the OT barycenter problem and express the OT cost with Monge's formulation:

$$\mathcal{L}^* \triangleq \sum_{k=0}^{K} \lambda_k \mathrm{OT}_{C_k}(\mathbb{P}_k, \mathbb{Q}^*). \tag{25}$$

where $\mathbb{Q}^*$ is the true barycenter distribution. By introducing the Monge's OT formulation with the true OT map $T^*$ that satisfies $T^*_\# \mathbb{P}_0 = \mathbb{Q}^*$, the expression can be rewritten as:

$$\mathcal{L}^* = \sum_{k=0}^{K} \lambda_k \int_{\mathcal{M}_k} C_k(\boldsymbol{m}_k, T^*(\boldsymbol{m}_0)) d\mathbb{P}_k(\boldsymbol{m}_k). \tag{26}$$

Due to the congruence condition on the potentials $\widehat{f}_{0:K}$ and the property $T^*_\# \mathbb{P}_0 = \mathbb{Q}^*$ for all $k$, we have:

$$\sum_{k=0}^{K} \lambda_k \mathbb{E}_{\boldsymbol{m}_k \sim \mathbb{P}_k}[\widehat{f}_k(T^*(\boldsymbol{m}_0))] = \mathbb{E}_{\boldsymbol{b} \sim \mathbb{Q}^*} \left[ \sum_{k=0}^{K} \lambda_k \widehat{f}_k(\boldsymbol{b}) \right] = 0. \tag{27}$$

This allows us to reformulate the optimal value $\mathcal{L}^*$. Using the definition of $\mathcal{F}$ in (20), we find:

$$\mathcal{L}^* = \sum_{k=0}^{K} \lambda_k \mathbb{E}_{\boldsymbol{m}_k \sim \mathbb{P}_k}[C_k(\boldsymbol{m}_0, T^*(\boldsymbol{m}_k))] - \underbrace{\mathbb{E}_{\boldsymbol{m}_{0:K} \sim \mathbb{P}_{0:K}} \left[ \sum_{k=0}^{K} \lambda_k \widehat{f}_k(T^*(\boldsymbol{m}_0)) \right]}_{=0 \text{ from } (27)}$$

$$= \mathbb{E}_{\boldsymbol{m}_{0:K} \sim \mathbb{P}_{0:K}} \left[ \sum_{k=0}^{K} \lambda_k \left( C_k(\boldsymbol{m}_k, T^*(\boldsymbol{m}_0)) - \widehat{f}_k(T^*(\boldsymbol{m}_0)) \right) \right] = \mathcal{F}(\widehat{f}_{0:K}, T^*).$$

With (23) we derive the second gap $\mathcal{E}_2$ can be written as

$$\mathcal{E}_2 = \mathcal{L}^* - \mathcal{L}(\widehat{f}_{0:K}) = \mathcal{F}(\widehat{f}_{0:K}, T^*) - \mathcal{F}(\widehat{f}_{0:K}, T^f). \tag{28}$$

We introduce the function $g_k(\boldsymbol{m}_k, \boldsymbol{b}) \triangleq C_k(\boldsymbol{m}_k, \boldsymbol{b}) - \widehat{f}_k(\boldsymbol{b})$, which is assumed to be $\beta$-strongly convex with respect to $\boldsymbol{b}$. Using this, we can rewrite our total cost functional $\mathcal{F}$ from (20) as:

$$\mathcal{F}(f_{0:K}, T) = \mathbb{E}_{\boldsymbol{m}_{0:K} \sim \mathbb{P}_{0:K}} \left[ \sum_{k=0}^{K} \lambda_k g_k(\boldsymbol{m}_k, T(\boldsymbol{m}_0)) \right]. \tag{29}$$

As a result of convexity, it follows that a necessary condition for $T^f$ to minimize $\mathcal{F}(f_{0:K}, T)$ is the vanishing of its first variation, yielding

$$\mathbb{E}_{\boldsymbol{m}_{0:K} \sim \mathbb{P}_{0:K}} \left[ \sum_{k=0}^{K} \lambda_k \nabla_{\boldsymbol{b}} g_k(\boldsymbol{m}_k, T^f(\boldsymbol{m}_0)) \right] = 0. \tag{30}$$

Now, we analyze the gap $\mathcal{E}_1$ by applying the $\beta$-strong convexity of $g_k(\boldsymbol{m}_k, \cdot)$:

$$\mathcal{E}_1 = \mathcal{F}(\widehat{f}_{0:K}, \widehat{T}) - \mathcal{F}(\widehat{f}_{0:K}, T^f)$$

$$= \mathbb{E}_{\boldsymbol{m}_{0:K} \sim \mathbb{P}_{0:K}} \left[ \sum_{k=0}^{K} \lambda_k \left( g_k(\boldsymbol{m}_k, \widehat{T}(\boldsymbol{m}_0)) - g_k(\boldsymbol{m}_k, T^f(\boldsymbol{m}_0)) \right) \right]$$

$$\geq \mathbb{E}_{\boldsymbol{m}_{0:K} \sim \mathbb{P}_{0:K}} \left[ \sum_{k=0}^{K} \lambda_k \left( \langle \nabla_{\boldsymbol{b}} g_k(\boldsymbol{m}_k, T^f(\boldsymbol{m}_0)), \widehat{T}(\boldsymbol{m}_0) - T^f(\boldsymbol{m}_0) \rangle + \frac{\beta}{2} \|\widehat{T}(\boldsymbol{m}_0) - T^f(\boldsymbol{m}_0)\|^2 \right) \right]$$

$$= \mathbb{E}_{\boldsymbol{m}_{0:K} \sim \mathbb{P}_{0:K}} \left[ \left\langle \sum_{k=0}^{K} \lambda_k \nabla_{\boldsymbol{b}} g_k(\boldsymbol{m}_k, T^f(\boldsymbol{m}_0)), \widehat{T}(\boldsymbol{m}_0) - T^f(\boldsymbol{m}_0) \right\rangle \right] + \frac{\beta}{2} \mathbb{E}_{\boldsymbol{m}_0 \sim \mathbb{P}_0} \left[ \|\widehat{T}(\boldsymbol{m}_0) - T^f(\boldsymbol{m}_0)\|^2 \right]$$

$$\overset{(30)}{=} 0 + \frac{\beta}{2} \mathbb{E}_{\boldsymbol{m}_0 \sim \mathbb{P}_0} \left[ \|\widehat{T}(\boldsymbol{m}_0) - T^f(\boldsymbol{m}_0)\|^2 \right]. \tag{31}$$

For the second gap $\mathcal{E}_2$, we conduct the same analysis and obtain

$$\mathcal{E}_2 \geq \frac{\beta}{2} \mathbb{E}_{\boldsymbol{m}_0 \sim \mathbb{P}_0} \left[ \|T^f(\boldsymbol{m}_0) - T^*(\boldsymbol{m}_0)\|^2 \right]. \tag{32}$$

Now we sum the inequalities for $\mathcal{E}_1$ (31) and $\mathcal{E}_2$ (32):

$$\mathcal{E}_1 + \mathcal{E}_2 \geq \frac{\beta}{2} \mathbb{E}_{\boldsymbol{m}_0 \sim \mathbb{P}_0} \left[ \|\widehat{T}(\boldsymbol{m}_0) - T^f(\boldsymbol{m}_0)\|^2 \right] + \frac{\beta}{2} \mathbb{E}_{\boldsymbol{m}_0 \sim \mathbb{P}_0} \left[ \|T^f(\boldsymbol{m}_0) - T^*(\boldsymbol{m}_0)\|^2 \right]$$

$$= \frac{\beta}{2} \mathbb{E}_{\boldsymbol{m}_0 \sim \mathbb{P}_0} \left[ \|\widehat{T}(\boldsymbol{m}_0) - T^f(\boldsymbol{m}_0)\|^2 + \|T^f(\boldsymbol{m}_0) - T^*(\boldsymbol{m}_0)\|^2 \right]$$

$$\geq \frac{\beta}{4} \mathbb{E}_{\boldsymbol{m}_0 \sim \mathbb{P}_0} \left[ \|\widehat{T}(\boldsymbol{m}_0) - T^*(\boldsymbol{m}_0)\|^2 \right] = \frac{\beta}{4} W_2^2(\widehat{T}_\# \mathbb{P}_0, T_\#^* \mathbb{P}_0) = \frac{\beta}{4} W_2^2(\widehat{T}_\# \mathbb{P}_0, \mathbb{Q}^*). \tag{33}$$

$\square$

### A.3 ANALYSIS ON BARYCENTER POLYTOPE VOLUME

Given the matrix containing vectors spanning the barycenter polytope

$$\mathbf{R} = (\boldsymbol{b}, \boldsymbol{r}_1, \cdots, \boldsymbol{r}_K),$$

we discuss the geometric meaning of barycenter polytope volumes for $n = 2$ and $n = 3$ modalities. $\langle \cdot, \cdot \rangle$ denotes the cosine similarity between two normalized vectors, so that

$$\langle \boldsymbol{b}, \boldsymbol{b} \rangle = \langle \boldsymbol{r}_1, \boldsymbol{r}_1 \rangle = \langle \boldsymbol{r}_2, \boldsymbol{r}_2 \rangle = 1. \tag{34}$$

We define the cosine of the angles between these vectors as

$$\cos \theta := \langle \boldsymbol{b}, \boldsymbol{r}_1 \rangle, \quad \cos \beta := \langle \boldsymbol{b}, \boldsymbol{r}_2 \rangle, \quad \cos \gamma := \langle \boldsymbol{r}_1, \boldsymbol{r}_2 \rangle. \tag{35}$$

For the special case $n = 2$, the squared volume can be expressed as the determinant

$$\mathrm{Vol}_{n=2}^2 = \det(\mathbf{R}^\top \mathbf{R}) = \begin{vmatrix} \langle \boldsymbol{b}, \boldsymbol{b} \rangle & \langle \boldsymbol{r}_1, \boldsymbol{b} \rangle \\ \langle \boldsymbol{b}, \boldsymbol{r}_1 \rangle & \langle \boldsymbol{r}_1, \boldsymbol{r}_1 \rangle \end{vmatrix} = 1 - \cos^2 \theta$$

Therefore, the polytope volume in the bimodal case reduces to:

$$\mathrm{Vol}_{n=2} = \sqrt{1 - \cos^2 \theta} = \sqrt{\sin^2 \theta} = \sin \theta,$$

which quantifies the spatial deviation between the WB embedding vector $\boldsymbol{b}$ and the vector $\boldsymbol{b} - \boldsymbol{m}_1$, reflecting how well the modality aligns with the overall barycenter direction. A smaller volume indicates stronger alignment, while a larger value implies greater directional discrepancy.

For the special case $n = 3$, the squared volume can be expressed as the determinant

$$\mathrm{Vol}_{n=3}^2 = \det(\mathbf{R}^\top \mathbf{R}) = \begin{vmatrix} \langle \boldsymbol{b}, \boldsymbol{b} \rangle & \langle \boldsymbol{r}_1, \boldsymbol{b} \rangle & \langle \boldsymbol{r}_2, \boldsymbol{b} \rangle \\ \langle \boldsymbol{b}, \boldsymbol{r}_1 \rangle & \langle \boldsymbol{r}_1, \boldsymbol{r}_1 \rangle & \langle \boldsymbol{r}_2, \boldsymbol{r}_1 \rangle \\ \langle \boldsymbol{b}, \boldsymbol{r}_2 \rangle & \langle \boldsymbol{r}_1, \boldsymbol{r}_2 \rangle & \langle \boldsymbol{r}_2, \boldsymbol{r}_2 \rangle \end{vmatrix}.$$

Substituting (34) and (35) into the determinant, the expression simplifies to

$$\mathrm{Vol}_{n=3}^2 = 1 - \cos^2 \theta - \cos^2 \beta - \cos^2 \gamma + 2 \cos \theta \cos \beta \cos \gamma,$$

$$= \sin^2 \theta + \sin^2 \beta + \sin^2 \gamma + 2 \cos \theta \cos \beta \cos \gamma - 2.$$

This form reveals the geometric interpretation more clearly: 1) The $\sin^2$ terms quantify the angular deviation of each vector pair. The $2 \cos \theta \cos \beta \cos \gamma$ term reflects the angular coupling between all three direction. 2) $\gamma$ is the angel between the two modality gap vectors $\boldsymbol{r}_1$ and $\boldsymbol{r}_2$, and thus directly reflects the structural inter-modality gaps between non-anchor modalities.

The volume becomes small when each modality vector closely follows the barycenter direction (small angles $\alpha$ and $\beta$). In addition, if the gap vectors between modalities, $r_1$ and $r_2$, are nearly aligned (small inter-modality angle $\gamma$), the volume is further reduced. In this case, all three angles are small, their sines are close to 0, and their cosines are close to 1. Conversely, the volume increases when the directions are mutually orthogonal, which maximizes the total dispersion among them. Fig. 6 plots the R@1 values of zero-shot retrieval on MSR-VTT with respect to the volume, showing a clear negative correlation between volume and performance.

Unlike pairwise cosine similarity, which only captures alignment between two modalities, the barycenter polytope volume offers a rich higher-order understanding of multimodal structure. It not only measures the holistic geometric alignment of each modality toward the barycenter, but also preserves inter-modal interactions, yielding a more compact multimodal joint representation space. In this sense, the volume serves as a non-trival metric for measuring $n$-modality alignment.

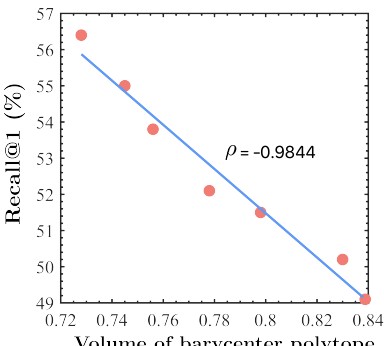

Figure 6: The polytope's volume is correlated ($\rho = -0.9844$) with the downstream performance.

# B EXPERIMENTAL DETAILS AND MORE RESULTS

## B.1 EXPERIMENTAL DETAILS

Table 5: Overview of multimodal benchmarks used for downstream evaluation.

| Benchmark | Modalities | Train | Val | Test | # Frames (train) | # Frames (test) |
|---|---|---|---|---|---|---|
| DiDeMo | Video + Text + Audio | 8,394 | 1,065 | 1,003 | 8 | 32 |
| ActivityNet | Video + Text + Audio | 10,009 | — | 4,917 | 8 | 32 |
| MSR-VTT | Video + Text + Audio + Subtitle | 9,000 | — | 1,000 | 8 | 8 |
| VATEX | Video + Text + Audio + Subtitle | 14,060 | — | 431 | 8 | 16 |
| VGGSound | Audio + Video + Text | — | — | 5000 | - | 8 |

Tab. 5 summarizes the modality configuration, data statistics, and frame settings across benchmarks.

**MSR-VTT** (Xu et al., 2016) is a widely used benchmark for video-text retrieval. It contains 10,000 video clips, each paired with approximately 20 textual captions, totaling around 200,000 captions. We use 9,000 videos for training and 1,000 for testing, with 8 frames sampled per video.

**DiDeMo** (Anne Hendricks et al., 2017) consists of 10,000 long-form videos, each annotated with 4 temporally ordered paragraph descriptions. It is mainly used for moment localization retrieval. The official split includes 8,394/1,065/1,003 videos for training/validation/testing, and 12 frames are sampled per video.

**ActivityNet** (Caba Heilbron et al., 2015) contains about 20,000 YouTube videos with a total duration of approximately 180 hours, annotated with multiple temporal sentence descriptions. We use the official training set (10,009 videos) for training and the validation set (4,917 videos) for downstream testing. The number of sampled frames per video is 8.

**VATEX** (Wang et al., 2019) includes around 25,000 English videos, each annotated with 10 English captions. It is commonly used for four-modality text retrieval tasks. We adopt 14,060 videos for training and 431 for testing, with 8 frames sampled per video.

**VGGSound5K** (Chen et al., 2020) is a 5,000-video subset of VGGSound, containing diverse audio event categories (typically 310 classes) along with corresponding video frames and subtitles. Each video clip in the dataset has a duration of 10 seconds and is annotated with a single label corresponding to the predominant sound event occurring within the clip. The dataset covers a wide spectrum of audio events, including human actions, animal vocalizations, natural phenomena, and mechanical sounds.It is commonly used for audio-video multimodal classification tasks.

## B.2 DISCUSSIONS ON THE ANCHOR SELECTION

To explore the impact of different anchor modalities to BaryBind, we conduct experiments using text and video as the original anchor sand evaluate on the VGGSound dataset (Chen et al., 2020) for video classification, and on MSR-VTT for V-T/T-V retrieval tasks. Note that the DAM loss is excluded in these settings, as it is inherently designed for text-based supervision.

Table 6: **Ablation on the anchor selection.**

| Anchor Modality | VGGSound | | | | MSR-VTT | | | |
| | Video | | Audio | | T2V | | V2T | |
| | Acc@1 | Acc@5 | Acc@1 | Acc@5 | R@1 | R@5 | R@1 | R@5 |
|---|---|---|---|---|---|---|---|---|
| Text Anchor | **47.8** | **74.2** | **45.2** | **72.4** | **53.2** | **77.0** | 50.6 | 73.8 |
| Video Anchor | 46.7 | 69.2 | 43.5 | 66.1 | 50.8 | 75.5 | **51.2** | **76.4** |

As shown in Tab. 6, the performance exhibits a slight decline when using the video modality as the anchor compared to the text modality. The result is consistent with the fact that text is often the most informative anchor in many multimodal tasks, as it retains the broad semantics of multimodal data.

## B.3 HYPERPARAMETER SENSITIVITY ANALYSIS

We study the sensitivity of the loss weights in the total objective on the MSR-VTT validation set:

$$\mathcal{L} = \mathcal{L}_{\text{MWB}} + \alpha_1 \mathcal{L}_{\text{BVC}} + \alpha_2 \mathcal{L}_{\text{DAM}}. \tag{36}$$

As reported in Tab. 7, adjusting $\alpha_1$ on MSR-VTT shows that increasing the weight of the BVC loss gradually improves retrieval performance, and the best results are achieved when $\alpha_1 = 1$. A further increase leads to a performance decline, indicating that over-restricting barycentric geometry may weaken instance-level discrimination.

Table 7: Sensitivity analysis of $\alpha_1$ on MSR-VTT validation set.

| $\alpha_1$ | 0.1 | 0.5 | 1 | 2 | 3 |
|---|---|---|---|---|---|
| T2V (R@1) | 54.2 | 54.9 | **56.5** | 55.3 | 54.5 |
| V2T (R@1) | 55.9 | 56.7 | **58.3** | 56.5 | 55.6 |

Table 8: Sensitivity analysis of $\alpha_2$ on MSR-VTT validation set.

| $\alpha_2$ | 0.02 | 0.05 | 0.1 | 0.15 | 0.2 |
|---|---|---|---|---|---|
| T2V (R@1) | 54.6 | 55.2 | **56.5** | 55.1 | 54.8 |
| V2T (R@1) | 55.2 | 55.8 | **58.3** | 56.2 | 55.1 |

Tab. 8 further reveals that $\alpha_2$ also has an optimal operating range, with $\alpha_2 = 0.1$ yielding the strongest retrieval performance. Excessively small or large weights cause suboptimal alignment due to either under-constrained or modality-biased instance matching.

Overall, $\alpha_1 = 1$ and $\alpha_2 = 0.1$ deliver the most stable and balanced multimodal alignment on the MSR-VTT validation set, demonstrating the complementary strengths of BVC and DAM.

## B.4 NORMALIZATION STRATEGIES FOR POLYTOPE VOLUME METRIC

While the raw barycenter polytope volume $V$ serves as a global alignment metric, it depends on the number of modalities and embedding dimensionality, which limits interpretability and cross-setting comparability. To address this, we consider normalization strategies such as $V^{1/n}$, which scale the volume to better reflect per-modality contributions and make the metric more comparable across different modality configurations.

Table 9 reports zero-shot retrieval results on MSR-VTT under different normalization strategies. We

Table 9: Zero-shot generalization on MSR-VTT under different normalization strategies for the barycenter polytope volume.

| Normalization | Setting | T2V | V2T |
|---|---|---|---|
| $V^{1/2}$ | T-V | 52.8 | 51.0 |
| $V^{1/3}$ | T-VA | 53.6 | 52.2 |
| $V^{1/4}$ | T-VAS | 55.4 | 52.7 |
| $V$ | T-V | 53.4 | 51.3 |
| $V$ | T-VA | 54.5 | 52.0 |
| $V$ | T-VAS | 56.3 | 53.6 |

observe that applying normalization slightly reduces
the absolute retrieval scores compared to the raw
volume $V$, likely because normalization reduces the
magnitude of the gradient signal from the volume metric, slightly weakening the alignment supervision. Nonetheless, the overall performance remains stable, indicating that $V^{1/n}$ provides a reliable and interpretable metric without significantly sacrificing retrieval accuracy.

## B.5 ROBUSTNESS TO MISSING MODALITIES

BaryBind effectively captures modality-agnostic semantics from arbitrary subsets of multimodal inputs, enabling the aligned representations of available modalities to serve as proxy features when others are missing. We evaluate two settings: (1) training-time missing-audio for cross-modal retrieval (T2A/A2T) on AudioCaps (Kim et al., 2019), and (2) missing-video inference for multimodal event classification on VGGSound 5K, where the model is trained with videos and audios. In both cases, the barycenter is optimized using only the accessible modalities. Results show that Bary-Bind preserves robust and graceful degradation under missing-modality conditions, which can be attributed to the barycenter modeling of modality-agnostic semantics.

Table 10: Text-to-audio retrieval on AudioCaps w/ and w/o audio during training.

| Training Setting | T2A R@1 | T2A R@10 |
|---|---|---|
| VAST (w/ audio) | 32.1 | 65.4 |
| GRAM (w/ audio) | 33.2 | 75.3 |
| BaryBind (w/ audio) | 35.5 | 81.2 |
| VAST (w/o audio) | 10.4 | 32.8 |
| GRAM (w/o audio) | 12.8 | 35.1 |
| BaryBind (w/o audio) | 21.1 | 56.2 |

Table 11: Multimodal event classification on VGGSound5K w/o video during inference.

| | Input modality | Acc@1 | Acc@5 |
|---|---|---|---|
| VAST | A+V | 48.1 | 79.6 |
| GRAM | A+V | 42.3 | 74.5 |
| BaryBind | A+V | 55.6 | 83.4 |
| VAST | A | 40.8 | 71.6 |
| GRAM | A | 38.5 | 70.1 |
| BaryBind | A | 49.4 | 78.3 |

## B.6 SYSTEM EFFICIENCY COMPARISON DURING TRAINING

We provide a comparison of system efficiency among VAST, GRAM, and BaryBind. All experiments are conducted on 2×NVIDIA A100 80GB GPUs with mixed-precision (FP16/AMP).

Table 12: System efficiency comparison during training.

| Model | Params | Batch size | Forward+Backward | Steps/Epoch | Time/Epoch |
|---|---|---|---|---|---|
| VAST | 1.28B | 64 | ~8.8s | 2344 | ~5.7h |
| GRAM | 1.30B | 64 | ~9.4s | 2344 | ~6.1h |
| BaryBind | 1.34B | 64 | ~9.7s | 2344 | ~6.3h |

The additional computation in BaryBind mainly comes from the Wasserstein barycenter optimization and auxiliary alignment losses. The increase in per-step time remains moderate while providing improved multimodal alignment performance.

## B.7 MORE QUANTITATIVE RESULTS ON SCALING TO MORE MODALITIES

To systematically evaluate the scalability of multimodal models with respect to the number of input modalities, we conduct experiments under four configurations, progressively increasing from two modalities (text and video) to five (text, video, audio, subtitle, and depth). This stepwise setup enables a controlled analysis of how each additional modality affects performance and alignment quality. For consistency and interpretability, text is used as the anchor modality throughout.

As shown in Table 13, both VAST and BaryBind benefit from the inclusion of additional modalities, demonstrating improved performance on MSR-VTT and VATEX in terms of both T2V and V2T retrieval. Notably, BaryBind consistently outperforms VAST across all modality configurations and datasets, highlighting its stronger scalability and generalization capacity.

Table 13: **Downstream performance with increasing number of modalities.**

| Modalities | | | | | MSR-VTT | | | | VATEX | | | |
|---|---|---|---|---|---|---|---|---|---|---|---|---|
| | | | | | VAST | | BaryBind | | VAST | | BaryBind | |
| Text | Video | Audio | Sub. | Depth | T2V | V2T | T2V | V2T | T2V | V2T | T2V | V2T |
| ✓ | ✓ | ✗ | ✗ | ✗ | 48.7 | 43.2 | 53.4 | 51.3 | 78.8 | 77.0 | 82.3 | 79.8 |
| ✓ | ✓ | ✓ | ✗ | ✗ | 49.3 | 43.7 | 54.5 | 52.0 | 80.0 | 77.3 | 84.2 | 81.3 |
| ✓ | ✓ | ✓ | ✓ | ✗ | 50.9 | 47.9 | 56.3 | 53.6 | 82.1 | 78.7 | 84.6 | 83.5 |
| ✓ | ✓ | ✓ | ✓ | ✓ | 51.2 | 49.3 | 57.0 | 54.4 | 82.4 | 79.2 | 84.9 | 83.8 |

## B.8 VISUALIZATION OF TOP-1 RETRIEVAL RESULTS

To qualitatively assess the retrieval performance of BaryBind, we visualize top-1 text-to-video results compared with two strong baselines in Fig. 8. BaryBind integrates text, audio, video, and subtitle modalities during both training and inference. Each row shows five frames from the top-retrieved video along with the query subtitle. In the first example, BaryBind retrieves a beach party scene aligned with the query's semantic and acoustic mood, while baselines return less relevant results. In the second case, BaryBind accurately matches a gameplay scene described by both visual and subtitle cues, whereas baselines retrieve unrelated content. The third example features a simple emotional phrase, "I'm scared.", where BaryBind selects an animated video of a fearful cat-dog chase, while others fail to reflect the emotional context or core entities. These results demonstrate BaryBind's ability to leverage complementary multimodal cues for precise and context-aware retrieval, effectively grounding both semantics and affect across diverse scenarios.

## B.9 SCALABILITY OF BARYBIND WITH INCREASING MODALITY NUMBER

Table 14: Computation time (in seconds) of similarity metrics vs. number of modalities, measured on an NVIDIA A100 GPU with batch size $B = 64$ and embedding dimension $D = 512$.

| Number of modality $n$ | 2 | 3 | 4 | 5 | 10 | 20 |
|---|---|---|---|---|---|---|
| Pairwise cosine similarity | $3.0 \times 10^{-7}$ | $7.0 \times 10^{-7}$ | $1.0 \times 10^{-6}$ | $1.3 \times 10^{-6}$ | $2.9 \times 10^{-6}$ | $4.9 \times 10^{-6}$ |
| Barycenter polytope volume | $4.9 \times 10^{-6}$ | $6.8 \times 10^{-6}$ | $5.9 \times 10^{-6}$ | $9.8 \times 10^{-6}$ | $3.6 \times 10^{-5}$ | $8.8 \times 10^{-5}$ |

We evaluate the computation efficiency of different similarity metrics with varying modality counts $n$. For each metric, we randomly sample $B = 64$ sets of $n$ vectors in $\mathbb{R}^D$ with $D = 512$, simulating multimodal embeddings. As shown in Tab. 14, the barycenter polytope volume remains computationally efficient and scales reasonably as the number of modalities increases. The negligible overhead incurred when extending to more modalities highlights the polytope volume as a nontrivial and scalable metric for assessing $n$-modality alignment.

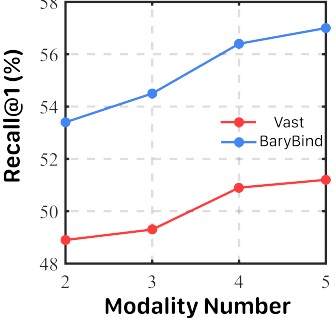

Figure 7: Zero-shot text-to-video retrieval results as scaling from 2 (T-V) to 5 (T-VASD) modalities.

To evaluate how BaryBind scales from bimodal (e.g., text-vision) setups to richer multimodal settings, we progressively expand the input from a basic text-video (T-V) pair to more complex configurations on the MSR-VTT dataset: text-video-audio (T-VA), text-video-audio-subtitle (T-VAS), and finally text-video-audio-subtitle-depth (T-VASD). The depth modality is derived using ChronoDepth (Shao et al., 2025) and integrated via an additional lightweight head attached to the vision encoder. As shown in Fig. 7, BaryBind consistently improves Recall@1 as the number of modalities increases, significantly outperforming the VAST baseline on MSR-VTT across all configurations. This highlights the scalability of BaryBind in practical multimodal understanding as it effectively integrates more modalities to shape a richer semantic space.

## Use of Large Language Models

We acknowledge that Large Language Models (LLMs) were used after the completion of the draft, solely to correct grammar and improve sentence fluency.

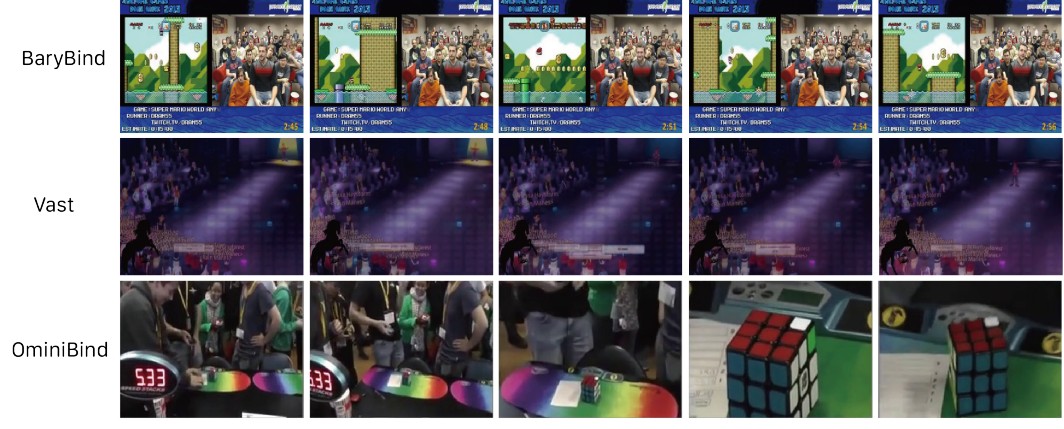

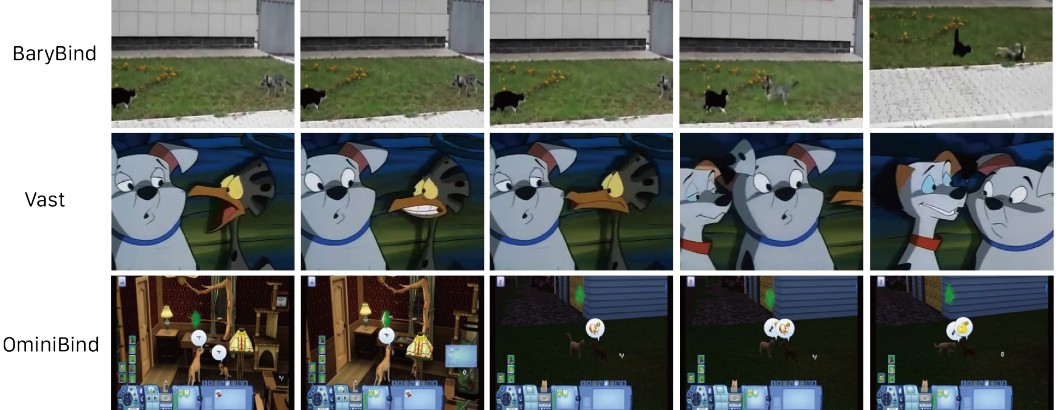

Figure 8: Visual results of text-to-video retrieval. We display 5 frames from the top-1 video.