# OpenReview forum: "BaryBind: Binding All Modalities via Multimodal Wasserstein Barycenter Space"
_ICLR.cc/2026/Conference — ICLR 2026 Conference Withdrawn Submission_

### Official Review · Reviewer_kVFT · 2025-10-24

**Soundness:** 3
**Presentation:** 3
**Contribution:** 2
**Rating:** 4
**Confidence:** 3

**Summary:**

This paper aims to generate multimodal representations by aligning different modalities into a common latent space. The proposed method is evaluated on text-to-video (T-to-V), video-to-text (V-to-T), multimodal classification, and retrieval tasks, demonstrating promising performance gains over existing methods such as VAST.

**Strengths:**

- The authors formulate the problem within a multimodal learning framework that aligns diverse modalities in the Wasserstein barycenter (WB) space and propose three complementary loss functions to effectively optimize the network.

- The method is evaluated on multiple benchmark datasets, showing consistent and competitive performance.

- Comprehensive ablation studies are conducted to assess the contribution of individual components (e.g., each loss term) and to analyze the effect of different experimental settings such as varying the anchor modality.

**Weaknesses:**

- The authors claim that existing models typically fix a specific modality (e.g., text) as the alignment anchor to bind others via pairwise contrastive losses, which limits scalability beyond two modalities. Although BaryBind aims to align all modalities to a shared latent space via distribution matching based on the Wasserstein distance, it still appears to treat the text modality as the anchor through simple MLP mappings. Consequently, the proposed method may still be minimizing the discrepancy primarily between the text modality and others. It is unclear to what extent, BaryBind differs from the existing solutions to address the anchoring issue highlighted in the abstract.

- Missing evaluation on system efficiency during training: The reported experiments were conducted on two NVIDIA A100 GPUs. It would be helpful to provide a direct comparison with existing model to illustrate the trade-off between efficiency and effectiveness.

- The proposed method is not evaluated on text-to-audio (T-to-A) or audio captioning tasks. Since most experiments focus on T-to-V and V-to-T tasks, it would strengthen the paper to demonstrate that the approach generalizes across modalities by including results on text–audio datasets such as Clotho and AudioCaps. Otherwise, it remains unclear whether the generated representations are biased toward certain modalities.

- Some important related works are missing and should be discussed for completeness:

Explore the Limits of Omni-modal Pretraining at Scale, ICCV 2025.

ViT-Lens: Towards Omni-modal Representations, CVPR 2024.

**Questions:**

Some additional questions are given below:

- Is the dataset used for training the proposed method identical to that used in VAST? It appears that the current work employs VAST-150K, a subset of VAST-27M. Please clarify the rationale behind choosing this subset.

- It would be beneficial to include a comparison against recent open-source multimodal large language models (e.g., Qwen, Qwen-Audio) to contextualize the proposed model’s performance relative to emerging large-scale multimodal baselines.

---

> ### Author Response · Authors · 2025-11-25
>
> Dear Reviewer kVFT,
>
> Thank you for your encouraging feedback! We are delighted that you found our framework effective and our evaluation comprehensive. Here are our point-by-point responses to your concerns:
>
> **Q1. Clarification on the difference from traditional anchor-based methods.**
>
> We thank the Reviewer for this insightful question, as it touches the core contribution of our work. Existing approaches rely on a fixed modality (typically text) as the semantic anchor, which inherently carries modality-specific bias into the joint space. Our design is intended precisely to overcome this issue.
>
> While our framework computationally begins from a specific modality anchor (text), our primary goal is to use the Wasserstein Barycenter (WB) formulation (Eq. 5) to **actively eliminate the modality bias by establishing WB embedding as the new anchor**. This is achieved by using a lightweight MLP trained with the MWB loss, which transforms the initial embedding towards a modality-agnostic anchor, i.e., the Wasserstein Barycenter. **Crucially, during optimization, all modalities jointly update the barycenter space, ensuring that no single modality (including text) dominates or serves as a fixed semantic anchor even though the MLP is fed with the initial anchor.**  Once the BVC loss is employed to align the non-anchor modalities to the WB for holistic multimodal binding.
>
> This mechanism is empirically validated in Fig. 2, in which the WB-based alignment leads to a well-balanced clustering around the barycenter and reduces modality dominance in the shared representation space.
>
> **In response to your valuable feedback, we have revised the manuscript to better emphasize the contributions of BaryBind, particularly its advantages/differences over traditional anchor-based methods.**
>
> **Q2. System efficiency comparison during training**
>
> Thanks for the valuable suggestion! We have provided the system efficiency comparison below. All experiments are conducted on 2×NVIDIA A100 80GB GPUs using mixed-precision (FP16/AMP).
>
> Table r4-1. System efficiency comparison during training.
>
> | Model      | Params | Batch size | Forward+Backward per step | Total steps per epoch | Total training time per epoch |
> |-----------|--------|-----------|---------------------------|---------------------|------------------------------|
> | VAST      | 1.28B   | 64        | ~8.8 s                      | 2344             | ~5.7 h                        |
> | GRAM      | 1.3B   | 64        | ~9.4s                      | 2344             | ~6.1 h                        |
> | BaryBind  | 1.34B  | 64        | ~9.7 s                    | 2344            | ~6.3 h                     |
>
> BaryBind incurs slightly higher per-step time due to the Wasserstein Barycenter (WB) computation and additional auxiliary losses. Nevertheless, the increase in total training time is moderate and is justified by the corresponding improvements in multimodal alignment and cross-modal retrieval performance.
>
> We have added the efficiency comparison in Appendix B.6.
>
> **Q3. Evaluation on text-to-audio (T2A) tasks.**
>
> We thank the Reviewer for the constructive suggestion! To investigate the behavior of BaryBind under text-to-audio (T2A) tasks, we conduct two experiments on AudioCaps:
> * T-VA training setting: text, video, and audio are all included during training.
> * Audio-missing T-V training setting: only text and video modalities are used for training, while the model can still infer audio via the barycenter-aligned latent space.
>
> The results are summarized in Table r3-2.
>
> Table r4-2. Text-to-audio retrieval on AudioCaps w/ and w/o audio during training.
> || Training Setting             | T2A R@1 | T2A R@10 |
> |:---------------------------:|:-------:|:-------:|:-------:|
> | VAST (w/ audio)      | T-VA|32.1   | 65.4    |
> | GRAM (w/ audio)       |T-VA |33.2   | 75.3    |
> | BaryBind (w/ audio)   |T-VA |35.5   | 81.2    |
> | VAST (w/o audio)      |T-V | 10.4    | 32.8    |
> | GRAM (w/o audio)       |T-V | 12.8    | 35.1    |
> | BaryBind (w/o audio) |T-V |21.1    | 56.2    |
>
> These results show that BaryBind achieves the highest retrieval performance when evaluated on text-audio tasks, demonstrating that the barycenter effectively alleviates the modality bias issue of the unified representation space.
>
> We have included the evaluation results in Appendix B.5 to further support our claims and strengthen the paper. Thanks again for your constructive suggestions!

---

> ### Author Response · Authors · 2025-11-25
>
> **Q4. Discussions on related works**
>
> We have updated the Related Work section to more clearly situate our contributions within recent directions. MiCo (zhang2024explore) effectively expands modalities, data, and model size to learn unified joint representations, and ViT-Lens (lei2024vit) innovatively adapts a pretrained ViT via modality-specific “lenses” to establish a shared space.
>
> **Q5. Training details (on the VAST150K) and training from scratch evaluation**
>
> We thank the Reviewer for the insightful comment. Due to updated YouTube policies, the full VAST pretraining dataset is no longer fully accessible. To address this, we perform continued training from the VAST model with BaryBind’s losses for one epoch on a VAST subset (VAST-150K) to reshape the latent space. To ensure a fair comparison and better highlight the advantages of BaryBind, we train it from scratch on MSR-VTT and ActivityNet, following the ablation strategies of VAST. The R@1 scores are reported in the table below, and training dynamics are visualized in Figure 4.
>
>
> | Method   | MSR-VTT (T2V) | MSR-VTT (V2T) | ActivityNet (T2V) | ActivityNet (V2T) |
> |----------|:-------------:|:-------------:|:-----------------:|:-----------------:|
> | VAST     | 35.0          | 39.1          | 26.5              | 30.5              |
> | BaryBind | 41.4          | 42.9          | 30.8              | 32.4              |
>
> The proposed BaryBind model significantly improves the alignment of the entire latent space thus leading to improved performance with respect to VAST (in the Table) and to VAST trained with cosine similarities. Therefore, in a fair training comparison from scratch on the same dataset BaryBind outperforms the cosine-based VAST comparison, proving that the proposed loss functions are more effective in modeling the multimodal interactions with respect to the conventional cosine-based methods. We additionally highlight the details in Sec. 4.1 of the revised paper.
>
> **Q6. Discussions on MLLMs**
>
> We appreciate the Reviewer’s suggestion regarding comparison with recent open-source multimodal large language models such as Qwen. While Qwen-VL and related decoder-only architectures demonstrate strong multimodal understanding capabilities, their publicly available results mainly focus on generative tasks such as visual question answering (VQA) and image/video captioning, rather than discriminative retrieval (T2V/V2T) or classification benchmarks that align with our evaluation protocol. Moreover, their model design—being decoder-only and generation-oriented—is architecturally different from our encoder-based discriminative framework, which is closer in spirit to CLIP-style contrastive models. Therefore, a direct and fair comparison under the same setup is currently not feasible. Nevertheless, we consider this a valuable direction for future work and are interested in exploring how BaryBind could be integrated into large multimodal LLM frameworks to enhance their multimodal understanding capabilities.
>
> ***Hope our explanations and experiments can address your inquiries, and we are happy to discuss any remaining questions.***

---

> ### Author Response · Authors · 2025-11-28
>
> Dear Reviewer kVFT,
>
> We hope this message finds you well. Thank you again for the insightful comments and valuable suggestions you provided in your review. As the discussion deadline on December 3 is approaching, we would greatly appreciate it if you could take a moment to check whether our rebuttal has resolved your points. Your insights are valuable to us, and if anything remains unclear or if further clarification would be helpful, we would be more than happy to provide it.
>
> Thank you sincerely for your time and effort in reviewing our work.
>
> Best regards,
>
> The Authors

---

### Official Review · Reviewer_oa7m · 2025-10-26

**Soundness:** 2
**Presentation:** 3
**Contribution:** 2
**Rating:** 2
**Confidence:** 5

**Summary:**

This paper investigates multimodal alignment by leveraging Wasserstein barycenter concept in Optimal transport theory. The main idea of the proposed method, BaryBind, is matching the Barycenter space derived from multimodal distributions to encourage each modality embedding being aligned to barycenter. Experiments show that the proposed method outperforms baseline methods, showing possible potential to enhance the multimodal alignment method.

**Strengths:**

The paper is well written and organized. Leveraging Wasserstein barycenter space for aligning multimodal embeddings is interesting. The current experimental results are also interesting as only the proposed model with pretraining only one-epoch can surpass other baseline models.

**Weaknesses:**

While the approach that leveraging OT theory looks interesting, I do not see a clear problem statement that the authors tackle and the rationale behind of using Wasserstein barycenter, which makes it seem incremental. Moreover, I am doubt with the experiment settings if it is fair comparison and if it properly shows the effect of the proposed loss functions. Below, I list some of my questions regarding my concerns.

**Questions:**

1. First of all, what is the main challenge or limitations of existing methods? From my understanding, this works tries to make modality-agnostic multimodal alignment as the previous works use anchor. If so, the challenge needs to be formally analyzed.
1. Constructing Wasserstein barycenter space needs barycenter, where the paper uses anchor-modality embedding for constructing it. This does not match the claim "modality-agnostic" as the method somehow biased to anchor modality when it align multimodal embeddings.
2. Regarding the above, using Wasserstein barycenter space as a contrasting point seems just aligning pair-wise distance from barycenter (some embedding of anchor modality) and all other non-anchor modalities (from eq(5)). This should be clearly compared mathematically and empirically.
3. I do not see the clear reasoning to use Data-anchor matching (DAM) loss. The paper only states "To complement the barycenter-based alignment loss..." Where does the problem behind of "instance-level supervision that encourages the model to distinguish between matched and mismatched pairs" come from and why do we need this? This requires clear motivation and analysis.
4. It seems BaryBind requires full multimodal samples, meaning that we must have all samples for every modality, otherwise BaryBind does not work. This is very important in practice as such multimodal datasets are rare.
5. Are the baseline methods (e.g., imageBind, languageBind, VAST, etc) trained on the same pretraining dataset with one epoch? Have the author compared baseline methods with the same backbone, the same hyperparameter, but only different in loss function? To fair compare and claim the superiority of the proposed loss, all the experimental setup should be identical.
6. Time complexity is only studied between the similarity measure. As the BaryBind has other modules, total training time (e.g., batch completion time) should be analyzed too.

---

> ### Author Response · Authors · 2025-11-25
>
> Dear Reviewer oa7m,
>
> Thank you for your valuable and detailed comments. We apologize for any confusion and have carefully revised the paper to clarify the main challenges and motivation, better explain the proposed loss functions (MWB, BVC, DAM), provide clearer descriptions of the pretraining and experimental setup, and include additional analyses such as missing-modality experiments and efficiency comparisons.
>
> **Q1. The main challenges of existing methods and restatement of BaryBind's motivation**
>
> We apologize for any confusion caused. The challenges we are addressing are mainly two-fold:
>
> * **Modality-specific bias in traditional anchors**: Existing anchor-based methods rely on a specific modality as the anchor, which can introduce modality-specific biases. **Our goal** is to filter out these biases and establish a modality-agnostic anchor via the Wasserstein barycenter, thereby achieving more balanced multimodal alignment, as validated in our experiments.
>
> * **Aligning n-modal data while preserving holistic geometry**: Previous methods typically perform pairwise alignment, which may overlook inter-modality interactions among non-anchor modalities. We address this by introducing the **Barycenter-anchored Volumetric Contrastive (BVC) strategy**, which aligns all modalities around a common barycenter while maintaining the geometric relationships of n-modal data.
>
> We have improved our presentation and claims in the abstract and introduction to emphasize the motivation.
>
> **Q2. Further clarification on the MWB loss (Eq. 5)**
>
> We thank the reviewer for the comment and apologize for any confusion. Indeed, Eq.5 is utilized to optimize and obtain the Wasserstein barycenter (WB) embedding via the multimodal Wasserstein Barycenter loss rather than aligning pair-wise distance to the barycenter. The global alignment across modalities is then enforced by the barycenter-anchored volumetric contrastive loss in Eq. 7. Therefore, Eq. (5) serves to learn the WB embedding itself, rather than performing pairwise alignment.
>
> \begin{align}
> \mathcal L^*_{\mathrm{MWB}}(\omega_{0:K},\theta) \triangleq \max_{\omega_{0:K}} \min_{\theta} \sum_{k=0}^K \lambda_k \mathbb E_{ m_k \sim \mathbb P_k} \Big[ \\| m_k - T_\theta(m_0) \\| - f_{\omega_k}(T_\theta(m_0)) \Big].   ~~~(\text{Eq 5})
> \end{align}
>
> During this optimization, only the weights $\theta$ of $T_\theta$ are updated, ensuring that the process affects only the WB embedding.
> We have enhanced the analysis of each loss component to clarify its specific role in the revised paper.
>
> **Q3. Clarification on the DAM loss**
>
> We thank the reviewer for the comment. The barycenter-based loss enforces global alignment but does not indicate whether a specific anchor–non-anchor pair is actually matched, which is standard in this family of multimodal retrieval methods [1,2]. The DAM loss addresses this by predicting matched vs. mismatched pairs, providing fine-grained supervision that improves cross-modal retrieval performance. We have revised section 3.4 accordingly.
>
> [1] Junnan Li, et al. Align before fuse: Vision and language representation learning with momentum distillation, NeurIPS 2021.
>
> [2] Sihan Chen, et al. VAST: A vision-audio-subtitle-text omni-modality foundation model and dataset, NeurIPS 2023.

---

> ### Author Response · Authors · 2025-11-25
>
> **Q4. Clarification on the requirement of full multimodal samples.**
>
> We thank the reviewer for raising this important point. In fact, BaryBind does not require all modalities for every sample. The Wasserstein barycenter can be computed from any subset of 2 to n modalities, enabling the model to generate aligned representations even when some modalities are missing. This flexibility is one of the key advantages of our approach. We evaluate two settings with three modalities (text, video, audio): (1) text-to-audio retrieval (T2A/A2T) on AudioCaps with missing audio during training, where audio is absent during training but available at inference; and (2) multimodal event classification with missing video during inference, where audio serves as the proxy for video during inference. In both cases, the barycenter is optimized using the accessible modalities, ensuring the task remains feasible.
>
> Table r2-2. Text-to-audio retrieval on AudioCaps w/ and w/o audio during training.
> || Training Setting             | T2A R@1 | T2A R@10 |
> |:---------------------------:|:-------:|:-------:|:-------:|
> | VAST (w/ audio)      | T-VA|32.1   | 65.4    |
> | GRAM (w/ audio)       |T-VA |33.2   | 75.3    |
> | BaryBind (w/ audio)   |T-VA |35.5   | 81.2    |
> | VAST (w/o audio)      |T-V | 10.4    | 32.8    |
> | GRAM (w/o audio)       |T-V | 12.8    | 35.1    |
> | BaryBind (w/o audio) |T-V |21.1    | 56.2    |
>
> Table. r2-3.  Multimodal event classification on VGGSound5K w/ and w/o video during inference.
> || Inference setting  | Acc@1 | Acc@5 |
> |:---------------------------:|:--------:|:--------:|:---:|
> | VAST   |  Audio + Video  | 48.1    | 79.6 |
> | GRAM    | Audio + Video | 42.3     | 74.5 |
> | BaryBind |   Audio + Video     |55.6     | 83.4 |
> | VAST   |  Audio only   | 40.8    | 71.6 |
> | GRAM    | Audio only| 38.5     | 70.1 |
> | BaryBind |    Audio only      | 49.4     | 78.3 |
>
> Results show that BaryBind preserves robust and graceful degradation under missing-modality conditions, which can credit to the barycenter modeling of modality-agnostic semantics.
>
> **Q5. Misunderstanding regarding pretraining and baseline comparison**
>
> Sorry for the confusion. In fact, for the zero-shot retrieval evaluation, we perform continued training for one epoch on a subset of VAST27M (due to YouTube policy restrictions on accessing the full video dataset). The baselines are evaluated using the same backbone, hyperparameters, and training setup, differing only in the loss function. We have re-emphasized these training details as ''We adopt VAST as backbone, with BERT-B for text, BEATs for audio, and EVA-CLIP-ViT-G for visual encoding.'', which ensures consistent and fair evaluation (see Sec. 4.1).
>
> To further validate fairness, we have also conducted training-from-scratch experiments on MSR-VTT, which yield consistent results confirming the superiority of our proposed loss.
>
> | Method   | MSR-VTT (T2V) | MSR-VTT (V2T) | ActivityNet (T2V) | ActivityNet (V2T) |
> |----------|:-------------:|:-------------:|:-----------------:|:-----------------:|
> | VAST     | 35.0          | 39.1          | 26.5              | 30.5              |
> | BaryBind | 41.4          | 42.9          | 30.8              | 32.4              |
>
> **Q6. Total Training Time and Time Complexity**
>
> We thank the reviewer for noting the importance of reporting total training time.
>
> For our 1.3B-parameter model in a contrastive learning setting for 4 modalities (TVAS), using a 2 A100 80GB GPU with batch size 64 and mixed-precision (FP16/AMP), we estimate the following:
>
> - **Forward + backward pass per step:** ~9.6 seconds
> - **Iterations per epoch (~150K samples):** ~2344
> - **Total training time per epoch:** ~6.2 hours
>
> ***Hope our response can address your inquiries, and we are happy to discuss any remaining questions.***

---

> ### Author Response · Authors · 2025-11-28
>
> Dear Reviewer oa7m,
>
> We hope this message finds you well. Thank you again for the constructive comments you provided in your review. As the discussion deadline on December 3 is approaching, we would greatly appreciate it if you could take a moment to check whether our rebuttal has resolved your points. Your insights are valuable to us, and if anything remains unclear or if further clarification would be helpful, we would be more than happy to provide it.
>
> Thank you sincerely for your time and effort in reviewing our work.
>
> Best regards,
>
> The Authors

---

### Official Review · Reviewer_EnWo · 2025-10-30

**Soundness:** 2
**Presentation:** 3
**Contribution:** 3
**Rating:** 6
**Confidence:** 4

**Summary:**

This paper introduces a novel approach to multimodal representation learning—BaryBind, whose core idea lies in addressing the issue of imbalanced representation space caused by anchor modality bias in conventional methods. Unlike existing approaches that typically align modalities around a specific anchor modality, BaryBind unifies the representations of different modalities by aligning them into a shared Wasserstein barycenter space. This method innovatively leverages the Wasserstein barycenter as a modality-agnostic semantic center, thereby effectively capturing semantics common to all modalities and achieving more balanced and robust alignment of multimodal representations.

**Strengths:**

The paper demonstrates significant strengths across several key dimensions:

**Originality & Conceptual Innovation** - The core contribution lies in fundamentally rethinking multimodal alignment objectives. By proposing the Wasserstein barycenter as a modality-agnostic semantic center, it shifts the paradigm from point-based anchoring to distribution-centered alignment. This conceptual breakthrough is further enhanced by the introduction of the barycenter polytope volume as a geometric metric, which transforms abstract alignment notions into computable quantities that naturally capture higher-order interactions beyond pairwise similarities.

**Theoretical Rigor & Technical Foundation** - The work establishes solid theoretical grounding through the dual formulation derivation of the MWB loss (Proposition 1), demonstrating careful mathematical development rather than heuristic design.

**Experimental Validation** - The experimental design stands out for its comprehensive coverage and convincing demonstrations across multiple benchmarks. The evaluation strategy effectively substantiates the method's advantages while maintaining scientific rigor in comparisons with state-of-the-art approaches.

**Presentation & Clarity** - Despite the conceptual complexity, the paper maintains logical coherence and accessibility through well-structured exposition. The introduction successfully frames the limitations of existing approaches and the paper's contributions, while Figures 1 and 3 provide exceptional visual intuition for understanding the core workflow and methodological distinctions from baselines.

**Weaknesses:**

**Theoretical Limitations in Wasserstein Barycenter Approximation** - While theoretically grounded in optimal transport, the practical implementation relies on a lightweight MLP $T_\theta$ to approximate the mapping to WB space through dual formulation. This parametric approximation raises questions about whether the method truly learns a distribution minimizing Wasserstein distances to all modalities, or merely converges to point estimates $b$ that optimize the specific loss function. The discrepancy between the theoretical WB (a distribution) and the implemented point estimate deserves further validation.

**Geometric Metric Limitations** - The barycenter polytope volume $V$, though innovative as a global alignment measure, presents interpretability challenges. As a scalar quantity, it effectively indicates the degree of misalignment but cannot identify which specific modalities contribute to the problem. Furthermore, its dependence on the number of modalities $n$ limits cross-model comparability, undermining its potential as a universal metric. Normalization strategies such as $V^{1/n}$ could enhance its applicability across different modality configurations.

**Insufficient Experimental Validation** - The experimental scope doesn't fully support the claims of modality-agnostic representation and improved inter-modal interactions. Critical tests for robustness under modality absence (e.g., missing video data during inference) are lacking, which would powerfully demonstrate advantages over anchor-based methods. Additionally, direct evidence for enhanced non-anchor modality interactions remains limited - probing tasks or mutual information analysis between modalities like audio and video under BVC constraints would provide more convincing validation.

**Questions:**

**Comparative Analysis of WB Approximation Methods** - To address the theoretical concerns regarding Wasserstein barycenter approximation, future work should compare the current MLP-based approach with more advanced neural optimal transport mappings, such as those proposed in [Kolesov et al., 2024a] and [Tang et al., 2025]. This comparative evaluation would help validate whether different approximation techniques significantly impact final performance and provide insights into the trade-offs between computational efficiency and theoretical fidelity.

**Normalization Strategies for Polytope Volume Metric** - For the barycenter polytope volume to serve as a universal alignment metric, investigation into normalization methods is essential. Exploring geometric normalization factors like $V^{1/n}$ could enable meaningful comparisons across models with varying numbers of modalities $n$. Developing a standardized normalization approach would enhance the metric's practicality and interpretability in diverse multimodal learning scenarios.

**Enhanced Experimental Validation through Probing Tasks** - To substantiate claims of improved modality-agnostic representation and inter-modal interactions, future experiments should include targeted probing tasks. These could assess model robustness under modality conflicts (e.g., contradictory audio and video signals) or directly quantify cross-modal relationships through feature correlation analysis and mutual information measurements between modalities under BVC constraints.

**References**
[1] Kolesov et al. Estimating barycenters of distributions with neural optimal transport. arXiv:2402.03828 (2024a)
[2] Tang et al. Baryir: Learning multi-source unified representation in continuous barycenter space for generalizable all-in-one image restoration. arXiv:2505.21637 (2025)

---

> ### Author Response · Authors · 2025-11-25
>
> Dear Reviewer EnWo,
>
> Thank you for raising these inspiring directions for future experiments. Here, we provide our responses and corresponding experiments for the concerns on theoretical limitations, geometric metric normalization, comparisons with other WB approximation architectures, and missing-modality robustness.
>
> **Q1. On the Error Bound of the barycenter map approximation**
>
> Thanks for this insightful comment regarding the theoretical guarantees of the barycenter approximation.  The following theorem establishes a formal error bound, proving that the distribution generated by our learned map, $\widehat T_{\\#}\mathbb P_0$, remains provably close to the ideal barycenter distribution $\mathbb Q^*$.
>
> Thank you for this insightful comment. You correctly point out that the Wasserstein barycenter is a distribution, while our implementation learns a parametric map, raising the concern of collapsing to a point estimate. We have strengthened the theoretical justification by introducing a duality-gap-based error analysis (Section 3.2), which quantitatively guarantees that the induced distribution remains close to the true barycenter.
>
>
> Specifically, we derive a **generalization-style error decomposition**:
>
> **Error bounds.** We establish the error bounds for the map $T$ with the following  simplified notations:
>   \begin{align}\mathcal F(f\_{1:K},T):=\mathcal L\_{\mathrm{MWB}}(f\_{1:K},T),~~~
>     	\mathcal L(f\_{1:K}):=\inf\_{T:\mathcal M\rightarrow\mathcal M\_B}\mathcal F(f\_{1:K},T),~~~ \mathcal L^\*:=\mathcal L\_{\mathrm{MWB}}^\*. \end{align}
>
> > **Theorem (Error analysis via duality gaps for the estimated barycenter distribution).**
> >
> > Let $C\_k$ be any transport costs. Assume that the maps $\mathbf{b} \to C\_k(\mathbf{m}\_k, \mathbf{b}) - \widehat{f}\_k(\mathbf{b})$ are $\beta$-strongly convex for $\mathbf{m}\_k \in \mathcal{M}\_k$, $k \in \lbrace0, ..., K\rbrace$. Consider the duality gaps for an approximate solution $(\widehat f\_{0:K},\widehat T)$ of our optimization problem:
> >
> > $$
> > \begin{align}
> >    \mathcal{E}\_1(\widehat{f}\_{0:K}, \widehat{T}) &\triangleq \mathcal{F}(\widehat{f}\_{0:K}, \widehat{T}) - \mathcal{L}(\widehat{f}\_{0:K}); ~~~~~\mathcal{E}\_2(\widehat{f}\_{0:K}) \triangleq \mathcal{L}^\* - \mathcal{L}(\widehat{f}\_{0:K}),
> > \end{align}
> > $$
> >
> > which represent the suboptimality of the learned map and potentials. Then the following inequality holds:
> >
> > $$
> > W\_{2}^{2}\left(\widehat T\_{\\#}\mathbb P\_0, \mathbb Q^\*\right) \leq \frac{4}{\beta}(\mathcal E\_1+\mathcal E\_2).
> > $$
>
> The result provides a standard two-term decomposition of the barycenter approximation error:
>
> - **Estimation error** $\mathcal E_1$: the suboptimality of the learned map relative to the best solution *within* the model class, caused by finite training.
> - **Approximation error** $\mathcal E_2$: the gap between the parametric model class and the ideal OT potentials, reflecting model capacity.
>
> We have added this theorem in section 3.2. The detailed proof is provided in the **Appendix A.2**. This theorem ensures that the Wasserstein distance between the estimated distribution $\widehat{T}_{\\#}\mathbb{P}_0$ and the true barycenter $\mathbb{Q}^*$ is upper-bounded by the sum of these two errors. This establishes that, as both approximation and estimation errors decrease during training, the learned distribution converges toward the true WB in a distributional sense.
>
>
>  **Q2. Discussion and comparison with other Wasserstein Barycenter approximations.**
> We thank the reviewer for this insightful suggestion. While the cited works [Kolesov et al., 2024a; Tang et al., 2025] focus on approximating barycenter of image features, our method operates in a multimodal latent space, which makes a direct comparison challenging. Nonetheless, we have attempted to adapt the architectures proposed in Kolesov et al. (i.e., VAE-based mappings) and Tang et al. (i.e., Transformer block-based mapping) to our multimodal setting with the same output dimension.
>
> Preliminary experiments in Table r2-1 indicate that although the approach can be integrated, the performance gains are limited in our scenario, likely due to the higher dimensionality and modality heterogeneity of the latent space. Nevertheless, it does yield a slight improvement compared to the baseline without WB. We will highlight these observations in the revised manuscript and note that further exploration of more expressive neural OT architectures in multimodal latent space is an interesting direction for future work.
>
> **Table r2-1. Zero-shot generalization on MSR-VTT with different WB approximation.**
>
> | Setting       | Backbone | T2V  | V2T  |
> |---------------|----------|------|------|
> | w/o WB        | T-VAS    | 50.6 | 46.4 |
> | Kolesov's WB  | T-VAS    | 52.1 | 49.2 |
> | Tang's WB     | T-VAS    |53.4  | 50.2 |
> | Ours          | T-VAS    | 56.3 | 53.6 |

---

> ### Author Response · Authors · 2025-11-25
>
> **Q3. Normalization strategies for polytope volume metric**
>
> We thank the reviewer for this insightful comment regarding the normalization of the barycenter polytope volume. As noted, while the raw volume $V$ serves as a global alignment metric, it depends on the number of modalities and embedding dimensionality, which limits interpretability and cross-setting comparability. Following your suggestion, we consider normalization strategies such as $V^{1/n}$, which scale the volume to better reflect per-modality contributions and make the metric more comparable across different modality configurations.
>
> **Table r2-2. Zero-shot generalization on MSR-VTT w/ and w/o volume normalization**
>
> | Normalization strategies | Setting | T2V  | V2T  |
> |-------------------------|---------|------|------|
> | $V^{1/2}$               | T-V     | 52.8 | 51.0 |
> | $V^{1/3}$               | T-VA    | 53.6 | 52.2 |
> | $V^{1/4}$               | T-VAS   | 55.4 | 52.7 |
> | $V$                     | T-V     | 53.4 | 51.3 |
> | $V$                     | T-VA    | 54.5 | 52.0 |
> | $V$                     | T-VAS   | 56.3 | 53.6 |
>
> Table r2-2 reports zero-shot retrieval results on MSR-VTT under different normalization strategies. We observe that applying normalization slightly reduces the absolute retrieval scores compared to the raw volume $V$, likely because normalization reduces the magnitude of the gradient signal from the volume metric, slightly weakening the alignment supervision. Nonetheless, the overall performance remains stable, indicating that $V^{1/n}$ provides a reliable and interpretable metric without significantly sacrificing retrieval accuracy. In future work, we plan to further explore such normalization strategies and their impact on multimodal alignment.
>
>
> **Q4. Robustness to Missing Modalities**
>
> Thanks for raising this important aspect, which highlights one key advantage of the barycenter in BaryBind: it effectively captures modality-agnostic semantics from arbitrary subsets of multimodal inputs, enabling the aligned representations of available modalities to serve as proxy features when others are missing. We evaluate two settings with three modalities (text, video, audio): (1) text-to-audio retrieval (T2A/A2T) on AudioCaps with missing audio during training, where audio is absent during training but available at inference; and (2) multimodal event classification with missing video during inference, where audio serves as the proxy for video during inference. In both cases, the barycenter is optimized using the accessible modalities, ensuring the task remains feasible.
>
> Table r2-3. Text-to-audio retrieval on AudioCaps w/ and w/o audio during training.
> || Training setting             | T2A R@1 | T2A R@10 |
> |:---------------------------:|:-------:|:-------:|:-------:|
> | VAST (w/ audio)      | T-VA|32.1   | 65.4    |
> | GRAM (w/ audio)       |T-VA |33.2   | 75.3    |
> | BaryBind (w/ audio)   |T-VA |35.5   | 81.2    |
> | VAST (w/o audio)      |T-V | 10.4    | 32.8    |
> | GRAM (w/o audio)       |T-V | 12.8    | 35.1    |
> | BaryBind (w/o audio) |T-V |21.1    | 56.2    |
>
> Table. r2-4.  Multimodal event classification on VGGSound5K w/ and w/o video during inference.
> || Inference setting  | Acc@1 | Acc@5 |
> |:---------------------------:|:--------:|:--------:|:---:|
> | VAST   |  Audio + Video  | 48.1    | 79.6 |
> | GRAM    | Audio + Video | 42.3     | 74.5 |
> | BaryBind |   Audio + Video     |55.6     | 83.4 |
> | VAST   |  Audio only   | 40.8    | 71.6 |
> | GRAM    | Audio only| 38.5     | 70.1 |
> | BaryBind |    Audio only      | 49.4     | 78.3 |
>
> Results show that BaryBind preserves robust and graceful degradation under missing-modality conditions, which can be credited to the barycenter modeling of modality-agnostic semantics.
>
> ***Hope our response can address your inquiries, and we are happy to discuss any remaining questions.***

---

> ### Author Response · Authors · 2025-11-28
>
> Dear Reviewer EnWo,
>
> We hope this message finds you well. Thank you again for the insightful comments and forward-looking suggestions you provided in your review. As the discussion deadline on December 3 is approaching, we would greatly appreciate it if you could take a moment to check whether our rebuttal has resolved your points. Your insights are valuable to us, and if anything remains unclear or if further clarification would be helpful, we would be more than happy to provide it.
>
> Thank you sincerely for your time and effort in reviewing our work.
>
> Best regards,
>
> The Authors

---

### Official Review · Reviewer_XuQB · 2025-11-01

**Soundness:** 2
**Presentation:** 2
**Contribution:** 2
**Rating:** 4
**Confidence:** 4

**Summary:**

The paper introduces BaryBind, a new framework used for aligning multimodalities to a Wasserstein barycenter (WB) space, say, a distribution in latent space that minimizes the average Wasserstein distance to each modality’s latent distribution. In addition, the paper introduces a volumetric metric among the embeddings of the modalities around the barycenter. and proposed a volumetric contrastive‐style loss for ensuring tighter alignment and reducing inter‐modality gaps. Experimental results demonstrate that BaryBind achieves significant performance improvement over baselines on cross-modal retrieval and classification tasks, highlighting the effectiveness of the proposed approach.

**Strengths:**

- Clear and Well-Structured: The paper is well-organized, with detailed explanations of the preliminary, intuition, and methodology.

- Interesting Method: The use of optimal transport / Wasserstein barycenters as a latent‐space tool is theoretically interesting.

- Superiority in Alignment: The experimental results demonstrate that the proposed method achieves the best performance on the cross-modal retrieval and classification tasks compared to the baselines.

**Weaknesses:**

- Currently, the MWB, BVC, and DAM loss objective functions are equally weighted in the combined loss, but it remains unclear whether assigning different weights could lead to better performance. A study of this trade-off would provide deeper insight into the relative importance of multimodalities' alignment preferences.

- The paper does not include experimental comparisons with other recent multimodal alignment methods, such as TRIANGLE [1] and GRAM [2]. Including these baselines would provide a stronger empirical validation of BaryBind’s effectiveness.

- The paper would benefit from a more in-depth ablation analysis. While the authors provide clear theoretical intuition and a validation experiment for each proposed component, the empirical section lacks a deeper discussion and interpretation of how these components individually and collectively contribute to the overall performance.

- Figures showing toy embeddings (before/after alignment) would help in visualizing the effect of the volumetric loss. Such visualizations could help demonstrate how embeddings converge toward the Wasserstein barycenter, how inter-modality gaps are reduced, and whether the volumetric constraint indeed promotes tighter alignment.

[1] A TRIANGLE Enables Multimodal Alignment Beyond Cosine Similarity, NeurIPS 2025

[2] Gramian multimodal representation learning and alignment, ICLR 2025

**Questions:**

How robust is it to missing modalities, which are common in realistic data? Does the barycenter degrade gracefully if one modality is missing?

---

> ### Author Response · Authors · 2025-11-25
>
> Dear Reviewer XuQB,
>
> Thank you for the encouraging feedback! We are delighted that you found our core concept intuitive and well-motivated. Here are our point-by-point responses to your comments:
>
>
> **Q1. Hyperparameter sensitivity analysis**
>
> Thanks for the insightful comment. We have conducted a sensitivity analysis on the loss weights in the total objective
> $$\mathcal L=\mathcal L_{MWB} + \alpha_1\mathcal L_{BVC} +\alpha_2\mathcal L_{DAM}$$ and the selected values ($\alpha_1=1, \alpha_2=0.1$) are shown to deliver the best performance observed on the MSRVTT validation set.
>
> Table r1-1. Hyperparameter sensitivity analysis of $\alpha_1$ and $\alpha_2$.
> | $\alpha_1$         |0.1 |0.5             | 0.8| 1              |1.2|1.5         |2 |
> |:---------------:|:--------------:|:--------------:|:--------------:|:--------------:|:--------------:|:---------:|:---------:|
> | **T2V (Acc / R@1)** |54.2 |54.9         |  55.8| **56.5**           |56.2| 55.7       |55.3 |
> | **V2T (Acc / R@1)** |55.9 |56.7         | 57.6| **58.3**           | 58.3|57.4         |56.5 |
>
> | $\alpha_2$         |0.02 |0.05             |0.08| 0.1              |0.12| 0.14    |0.16|
> |:---------------:|:--------------:|:--------------:|:--------------:|:--------------:|:--------------:|:--------------:|:---------:|
> | **T2V (Acc / R@1)** |54.6 |55.2         | 56.0 | **56.5**           | 55.8|55.8       |55.3 |
> | **V2T (Acc / R@1)** |55.2 |55.8         | 57.9 | **58.3**           | 57.7|57.5        |56.2 |
>
>  The configuration $\alpha\_1=1$ and $\alpha\_2=0.1$ achieves the best overall results, demonstrating that a properly balanced combination of BVC and DAM effectively enables complementary multimodal alignment. The result has been added  in Appendix B.3.
>
> **Q2. Comparison with GRAM and TRIANGLE**
>
>  We have added comparisons against GRAM and TRIANGLE under our evaluation settings. Specifically, GRAM is reproduced using the same hardware configuration (GPUs), number of sampled frames, and batch size to ensure a fair comparison on multimodal classification (VGGSound) and retrieval benchmarks. For TRIANGLE (NeurIPS 2025), we report the official results from their paper since pretrained models are not publicly released and their retrieval configuration differs from ours. The updated results are summarized below:
>
> Table r1-2. Comparison of multimodal classification on VGGSound5K
> | Method       | Modality | Acc@1 | Acc@5 |
> |--------------|----------|-------|-------|
> | GRAM         | V        | 43.1  | 71.8  |
> | GRAM         | A+V      | 42.3  | 74.5  |
> | TRIANGLE     | A+V      | 44.8  | 80.0  |
> | BaryBind (Ours) | A      | 45.7  | 75.2  |
> | BaryBind (Ours) | V      | 48.3  | 76.4  |
> | BaryBind (Ours) | A+V    | **55.6** | **83.4** |
>
> Table r1-3. Comparison of zero-shot T2V and V2T retrieval.
> | Method           | Modality | MSR-VTT T2V | MSR-VTT V2T |  VATEX T2V | VATEX V2T |
> |------------------|----------|-------------|-------------|-----------|-----------|
> | GRAM             | T-VAS    | 54.2        | 51.6        | 83.2      | 81.9      |
> | BaryBind (Ours)  | T-VAS    | **56.3**    | **53.6**    | **84.6**  | **83.5**  |
>
> Both works are closely related to our approach, and we have also discussed them in the Related Work section.
>
>
> **Q3. Deeper discussion and interpretation of component contributions**
>
> Thanks for the valuable suggestion! We have strengthened the interpretations and discussions of each component as follows:
> > * The MWB (Multimodal Wasserstein Barycenter) loss constructs the barycenter to filter out modality-specific biases in the original anchor (e.g., text), resulting in a modality-agnostic WB anchor that serves as the new alignment center.
> > * The BVC (Barycenter Volume Contrastive) loss leverages the WB as the new anchor to pull non-anchor modality embeddings closer to the barycenter by contrasting the volume of barycenter polytoes. This alignment process encourages holistic alignment towards the WB and preserves inter-modal interactions among non-anchor modalities, yielding a more balanced and modality-agnostic representation space.
> > * The DAM (Data-Anchor Matching) loss predicts whether a pair of multimodal data is matched or not, improving the discrimination between truly matched and semantically similar but unmatched pairs.
> > * The MWB and BVC losses work together to build a unified representation space centered on the WB anchor and reduce inter-modality gaps, in which the MWB loss learns the barycenter solution, and the BVC loss aligns multimodal embeddings toward it. When combined with the DAM loss, the model achieves stronger discrimination of mismatched pairs and more robust and balanced multimodal alignment.
>
> We have also incorporated the enhanced analysis in Section 4.3.

---

> > ### Author Response · Authors · 2025-11-25
> >
> > **Q4. Visualization of multimodal embeddings before and after alignment**
> >
> > Thank you for this excellent suggestion! In the revised version, we have added a visualization (see Figure 5) to show the toy embeddings before and after alignment. As shown in Figure 5, the embeddings become increasingly aligned around the Wasserstein barycenter from the baselines w/o alignment (MWB only) to pairwise alignment (MWB + TV + TA), and finally our proposed volumetric alignment (MWB + BVC). The BVC loss substantially reduces inter-modality gaps and yields tighter clusters around the barycenter, confirming that the volumetric constraint indeed enforces consistent cross-modal alignment.
> >
> > **Q5. Robustness to missing-modality scenarios**
> >
> > Thanks for raising this important aspect, which highlights one key advantage of the barycenter in BaryBind: it effectively captures modality-agnostic semantics from arbitrary subsets of multimodal inputs, enabling the aligned representations of available modalities to serve as proxy features when others are missing. We evaluate two settings with three modalities (text, video, audio): (1) text-to-audio retrieval (T2A/A2T) on AudioCaps with missing audio during training, where audio is absent during training but available at inference; and (2) multimodal event classification with missing video during inference, where audio serves as the proxy for video during inference. In both cases, the barycenter is optimized using the accessible modalities, ensuring the task remains feasible.
> >
> > Table r1-4. Text-to-audio retrieval on AudioCaps w/ and w/o audio during training.
> > || Training setting             | T2A R@1 | T2A R@10 |
> > |:---------------------------:|:-------:|:-------:|:-------:|
> > | VAST (w/ audio)      | T-VA|32.1   | 65.4    |
> > | GRAM (w/ audio)       |T-VA |33.2   | 75.3    |
> > | BaryBind (w/ audio)   |T-VA |35.5   | 81.2    |
> > | VAST (w/o audio)      |T-V | 10.4    | 32.8    |
> > | GRAM (w/o audio)       |T-V | 12.8    | 35.1    |
> > | BaryBind (w/o audio) |T-V |21.1    | 56.2    |
> >
> > Table. r1-5.  Multimodal event classification on VGGSound5K w/ and w/o video during inference.
> > || Inference setting | Acc@1 | Acc@5 |
> > |:---------------------------:|:--------:|:--------:|:---:|
> > | VAST   |  Audio + Video  | 48.1    | 79.6 |
> > | GRAM    | Audio + Video | 42.3     | 74.5 |
> > | BaryBind |   Audio + Video     |55.6     | 83.4 |
> > | VAST   |  Audio only   | 40.8    | 71.6 |
> > | GRAM    | Audio only| 38.5     | 70.1 |
> > | BaryBind |    Audio only      | 49.4     | 78.3 |
> >
> > Results show that BaryBind preserves robust and graceful degradation under missing-modality conditions, which can be credited to the barycenter modeling of modality-agnostic semantics.
> >
> > ***Hope our explanation and experiments can address your inquiries, and we are happy to discuss any remaining questions.***

---

> ### Author Response · Authors · 2025-11-28
>
> Dear Reviewer XuQB,
>
> We hope this message finds you well. Thank you again for the in-depth and technically rigorous comments you provided in your review. As the discussion deadline on December 3 is approaching, we would greatly appreciate it if you could take a moment to check whether our rebuttal has resolved your points. Your insights are valuable to us, and if anything remains unclear or if further clarification would be helpful, we would be more than happy to provide it.
>
> Thank you sincerely for your time and effort in reviewing our work.
>
> Best regards,
>
> The Authors

---

### Author Response · Authors · 2025-11-25
**Global Response**

We sincerely thank the Area Chair for handling our paper and the reviewers for their valuable feedback and constructive suggestions.

In this paper, we present BaryBind that aligns multiple modalities to the Wasserstein barycenter with a volumetric loss for establishing a modality-agnostic multimodal semantic space, enabling scalable and balanced multimodal understanding. We are encouraged by the recognition regarding the **novelty** and **motivation** of leveraging the barycenter for multimodal alignment (Reviewers kVFT and oa7m, EnWo), highlighting our approach as a **conceptual breakthrough** (EnWo), the **presentation** (all reviewers), and the **comprehensive evaluation** (EnWo and kVFT).

In response to your concerns, we have included detailed clarifications, extended results, and further discussions/analyses. Below is a summary of our responses and the revisions we have made to the manuscript:

>1. Further clarification on the motivation for using the barycenter and its difference between traditional anchor-based methods

Existing methods often rely on (i) a modality-specific anchor (e.g., text), which inevitably introduces modality bias into the shared space, and (ii) pairwise contrastive alignment, which overlooks higher-order inter-modality interactions and the holistic geometry of multimodal data. These limitations collectively lead to suboptimal and imbalanced multimodal representations.

 In contrast, we intend to mitigate this modality bias and establish a more modality-agnostic shared space via the Wasserstein barycenter (WB). Specifically,  A lightweight MLP trained with **the MWB loss** transforms the modality-specific anchor towards a more modality-agnostic anchor, i.e., the WB embeddings. **Crucially, during optimization, all modalities jointly update the barycenter space, ensuring that no single modality (including text) dominates or serves as a fixed semantic anchor even though the MLP is fed with the initial anchor.**  Once the WB is established, **the BVC loss** is employed to align the non-anchor modalities to the WB for multimodal binding, enabling more balanced and holistic multimodal alignment.

We have updated the abstract and introduction to better highlight the motivation and contributions of BaryBind.

>2. Clarification on pretraining details and dataset usage.

We have re-clarified that, due to YouTube policy restrictions on accessing the full video dataset, we perform continued training based on the VAST model on the VAST-150K dataset for the zero-shot cross-modal retrieval evaluation. In addition, we include training-from-scratch experiments and display the training dynamics (Section 4.3) on MSR-VTT and ActivityNet. All baselines are evaluated under the same setup, ensuring fair comparison and reproducibility.

We have further refined the implementation details and incorporated additional results in Section 4 to improve clarity and completeness.

>3. Error bound of the learned barycenter map

We provide a theoretical guarantee for the learned barycenter map: the Wasserstein distance between the induced distribution $\widehat T_{\\#}\mathbb P_0$ and the true barycenter $\mathbb Q^*$ is upper-bounded by the sum of the estimation error (due to finite training) and the approximation error (due to model capacity).

The theorem has been added as Theorem 3.1, and its proof is given in Appendix A.2.

>4. Additional experimental validation.

We have added extensive experimental results, including hyperparameter sensitivity studies (Appendix B.3), ablation analysis of MWB/BVC/DAM's contributions (Section 4.3), t-SNE visualizations of embedding alignment (Section 4.3), robustness under missing modalities (Appendix B.5), and so on. We also include comparisons with the latest multimodal baselines and other WB approximation architectures, evaluation on text-to-audio tasks, and provide system efficiency metrics. These experiments confirm that the barycenter serves as a more modalit-agnostic anchor for binding multiple representations, and maintains graceful performance degradation when some modalities are missing.

>5. Discussions on related works

We have updated the Related Work section to clearly situate our contributions, including MiCo, ViT-Lens, and TRIANGLE, highlighting how BaryBind’s barycenter-based approach improves multimodal alignment beyond conventional anchor-based or pairwise methods.

***The revised paper has been uploaded. We hope that these modifications help to strengthen the manuscript and align it more closely with your expectations.***

---

### Note · Authors · 2026-01-26

I have read and agree with the venue's withdrawal policy on behalf of myself and my co-authors.

---

### Meta-Review · Area_Chair_5MvW · 2026-01-04

**Summary:**

The reviewers identified several critical hurdles regarding the motivation, theoretical grounding, and evaluation of BaryBind. While the paper proposes using a Wasserstein Barycenter (WB) as a semantic center to avoid the bias of fixed-modality anchors (usually text), the technical execution relies on a transformation of the text modality itself to define that center.

The recommendation for Rejection is driven by the failure of the authors to resolve the "anchor bias" concern, a lack of clarity in responding to the most critical technical questions from Reviewer `oa7m`, and the remaining limitations of the proposed geometric metric identified by Reviewer `EnWo`.

**Reviewer Concerns:**

Concerns Addressed by the Rebuttal

* Missing baselines and ablations (`XuQB`, `kVFT`): The authors successfully added comparisons against recent state-of-the-art methods GRAM and TRIANGLE, and provided a hyperparameter sensitivity analysis for loss weights.

* Robustness to missing modalities (`EnWo`, `XuQB`): New experiments on AudioCaps and VGGSound (Tables r1-4, r1-5) demonstrate that the model handles missing modalities with graceful degradation, showing a clear empirical advantage over the VAST baseline.

* Efficiency (`kVFT`): The authors provided a wall-clock time comparison (Table r4-1) showing that BaryBind incurs a manageable ~10% overhead compared to standard models.

Outstanding Concerns

* Unsubstantiated "modality-agnostic" claim (`oa7m`, `kVFT`): This is the most significant unresolved issue. Reviewer `oa7m` noted that constructing the space using anchor-modality (text) embeddings is biased and invalidates the paper's core claim. The rebuttal (to `kVFT`) claimed:
> _"While our framework computationally begins from a specific modality anchor (text), our primary goal is to use the WB formulation to actively eliminate the modality bias by establishing WB embedding as the new anchor... ensuring that no single modality (including text) dominates."_

  This remains a circular justification. Transforming a text embedding may not completely remove the semantic bias of the text; it may still simply project it. The "barycenter" is technically a function of the text seed, making the claim of being "modality-agnostic" misleading. The authors' rebuttal to `oa7m`'s Q2 was particularly problematic, as it skipped the logic of the question entirely to address a different point (pairwise distance).

* Geometric metric limitations (`EnWo`): The "polytope volume" metric depends on the number of modalities $n$. Reviewer `EnWo` noted:
> "its dependence on the number of modalities $n$ limits cross-model comparability, undermining its potential as a universal metric."

  The rebuttal attempted a normalization strategy (Table r2-2) which reduced absolute retrieval scores and failed to demonstrate that the metric is truly interpretable across different modality configurations.

**Reviewer Scores:**

* Reviewer `oa7m` (Initial: 2 $\rightarrow$ Estimated: 2): The rebuttal was evasive on the core critique of text-seeding. The reviewer's high confidence (5/5) and the authors' failure to address the initialization bias mean this score would not have improved.

* Reviewer `XuQB` (Initial: 4 $\rightarrow$ Estimated: 5): While additional experiments were provided, the broader conceptual flaws regarding the "agnostic" claim would likely have prevented a full move to 6.

* Reviewer `EnWo` (Initial: 6 $\rightarrow$ Estimated: 4/5): The initial 6 appears inconsistent with the reviewer's own "Soundness: 2" rating. Given that the normalization strategy failed to improve results and the "point estimate vs. distribution" concern remains practically relevant, this reviewer would likely have downgraded.

* Reviewer `kVFT` (Initial: 4 $\rightarrow$ Estimated: 4): The reviewer’s primary concern about the "anchoring issue" was shared with `oa7m` and remained technically unresolved by the authors' circular explanation.

---

### Decision · Program_Chairs · 2026-01-26

Reject